# Experimental demonstration of tunable hybrid improper ferroelectricity in double-perovskite superlattice films

Yaoxiang Jiang [1,4], Jianguo Niu [1,4], Cong Wang [2,4] ✉, Donglai Xue[1], Xiaohui Shi[1], Weibo Gao [3] ✉ & Shifeng Zhao [1] ✉

Hybrid improper ferroelectricity can effectively avoid the intrinsic chemical incompatibility of electronic mechanism for multiferroics. Perovskite super-lattices, as theoretically proposed hybrid improper ferroelectrics with simple structure and high technological compatibility, are conducive to device integration and miniaturization, but the experimental realization remains elusive. Here, we report a strain-driven oxygen octahedral distortion strategy for hybrid improper ferroelectricity in $La_2NiMnO_6/La_2CoMnO_6$ double-perovskite superlattices. The epitaxial growth mode with mixed crystalline orientations maintains a large strain transfer distance more than 90 nm in the superlattice films with lattice mismatch less than 1%. Such epitaxial strain permits sustainable long-range modulation of oxygen octahedral rotation and tilting, thereby inducing and regulating hybrid improper ferroelectricity. A robust room-temperature ferroelectricity with remnant polarization of ~ 0.16 $\mu C\ cm^{-2}$ and piezoelectric coefficient of 2.0 $pm\ V^{-1}$ is obtained, and the density functional theory calculations and Landau-Ginsburg-Devonshire theory reveal the constitutive correlations between ferroelectricity, octahedral distortions, and strain. This work addresses the gap in experimental studies of hybrid improper ferroelectricity for perovskite superlattices and provides a promising research platform and idea for designing and exploring hybrid improper ferroelectricity.

The electric polarization of perovskite ferroelectrics is typically derived from the second-order Jahn-Teller effect of closed-shell $B$-site cations with either $d^0$ or $ns^2$ electronic configurations that are incompatible with magnetism, hampering the development of multiferroic materials[1,2]. Hybrid improper ferroelectricity (HIF) originates from structural geometry effects featured by the trilinear coupling associated with two nonpolar octahedral distortions[3], which effectively avoids the intrinsic chemical incompatibility of electronic mechanism[4]. Over the past two decades, HIF has been predicted only in two kinds of

structures, i.e., layered perovskites (Ruddlesden-Popper or Dion-Jacobson phase)[5–8] and perovskite superlattices[9–11], by combining group-theoretical symmetry analysis and first-principle calculations.

So far, it has only been experimentally detected in very few bulk materials of $A_3B_2O_7$-type layered perovskites[1,2,12,13]. However, the hybrid improper mechanism in layered perovskites requires large $B$-site cations to stabilize the highly distorted perovskite frameworks[2], since the $BO_6$ octahedral rotations are often partly or completely suppressed in the $A_2B_2O_6$ perovskite layers interrupted by the $AO$ rock-salt

[1]Inner Mongolia Key Lab of Nanoscience and Nanotechnology & School of Physical Science and Technology, Inner Mongolia University, Hohhot, PR China. [2]College of Mathematics and Physics, Beijing University of Chemical Technology, Beijing, China. [3]Division of Physics and Applied Physics, School of Physical and Mathematical Sciences, Nanyang Technological University, Singapore, Singapore. [4]These authors contributed equally: Yaoxiang Jiang, Jianguo Niu, Cong Wang. ✉e-mail: wangcongphysics@mail.buct.edu.cn; wbgao@ntu.edu.sg; zhsf@imu.edu.cn

layers[5,7]. Such coherence length constraint on the polar distortion limits strong polar mode-strain coupling, then the bulk in-plane polarization can hardly be reoriented to an out-of-plane direction in thin films[5], contrary to the request of functional device applications. Moreover, the effective force-field control of HIF has still not been achieved in any layered perovskite oxides[1,10]. More importantly, it is difficult for layered perovskites to realize epitaxial growth of thin films because of their complex crystal structures and large lattice parameters, which hinder the integration and miniaturization of devices.

Perovskite superlattices, as another theoretical system for the prediction of HIF, can provide excellent structural compatibility for epitaxial single-crystal thin films grown on suitable substrates, and the strain tunability for octahedral distortions in successive perovskite layers. Recently, the double-perovskite bilayer systems with the typical octahedral rotation mode of a⁻a⁻c⁺ (in Glazer's notation) are theoretically expected to achieve HIF[9,10]. However, it has remained difficult to induce and clarify HIF in relevant experimental systems. On the one hand, the influencing factors caused by local non-uniform strains are difficult to exclude, including the flexoelectric effect and displacement-type ferroelectricity[14,15]. On the other hand, the incomplete oxidation state and anti-site disorder of the *B*-site transition metal cations usually disturb stable octahedral distortion modes[16,17]. Consequently, it is still urgent whether hybrid improper ferroelectricity can be induced in a double-perovskite superlattice system.

In general, there are two main factors on experiment that hinder the design of HIF in double-perovskite superlattices. Firstly, the growth window of *B*-site ordered double perovskites is quite narrow[18,19], and during the film growth, the incomplete oxidation state of transition metal cations inevitably leads to various lattice defects[20–23]. Thus, the fabrication of high-quality double-perovskite films with *B*-site ordering is a necessary and challenging prerequisite

for the identification of HIF. Secondly, strain-induced structural distortions are quite complex and difficult in tailoring[24]; and the epitaxial strain in thin films not only depends on the large lattice mismatch[25,26], but also decays rapidly with increasing film thickness[27,28]. Especially, once the lattice parameters of the substrates and thin films are determined for epitaxial growth, the magnitude of oxygen octahedral rotation and tilting (OOR and OOT) can no longer be flexibly regulated. These difficulties in tuning the epitaxial strain and oxygen octahedral distortion for perovskite superlattice systems hinder the triggering and realization of HIF in experiments. Therefore, designing and searching for experimental cases of perovskite superlattices with hybrid improper mechanisms has become an indispensable and urgent challenge on HIF research. Notably, for determining HIF, the thickness-dependent epitaxial strain can maintain a single mode of octahedral distortion during strain regulation, which facilitates the demonstration of non-monotonic ferroelectric contributions from the coupling of OOR and OOT.

In this work, we overcame the universal difficulties of the sustainable long-range modulation for octahedral distortion in designing HIF by regulating thickness-dependent epitaxial strain and achieved a tunable HIF at room temperature in La₂NiMnO₆/La₂CoMnO₆ double-perovskite superlattices. Although the obtained room-temperature HIF still have a non-negligible shortfall with conventional ferroelectrics[29–31], it is comparable to the current excellent magnetic unconventional ferroelectrics[32,33]. We utilized an ozone-assisted growth method to achieve the growth of high-quality double-perovskite superlattices with *B*-site ordering by optimizing the conditions of ozone concentration. A large strain critical thickness is achieved by the epitaxial growth mode with mixed crystalline orientations, which permits the sustainable long-range modulation of oxygen octahedral rotation and tilting. Such strain-driven octahedral distortion strategy

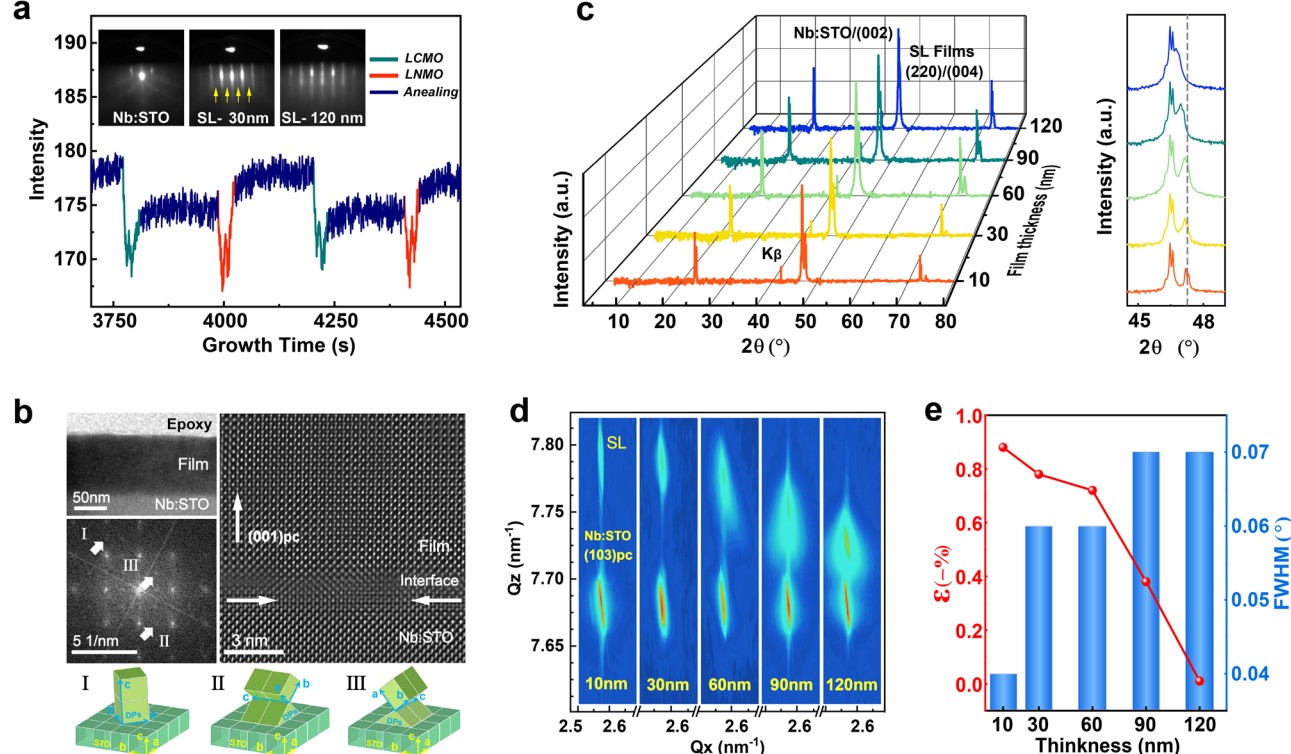

**Fig. 1 | Epitaxial growth and strain of La₂NiMnO₆/La₂CoMnO₆ superlattice films. a** Reflection high-energy electron diffraction (RHEED) patterns and intensity oscillations during the film growth. The twofold superstructure peaks are marked by the yellow arrows. **b** Cross-sectional TEM image of SL₉₀ at low magnification, and HRTEM image at the interface between the films and substrates. The additional diffraction spots numbered by white arrows in the FFT image

correspond to three growth modes of double perovskites. **c** X-ray diffraction θ-2θ scan of the superlattice films with different thicknesses grown on Nb:STO substrates. The inset is a magnification of the epitaxial peaks along (002)_Nb:STO peaks. **d** XRD reciprocal space maps around Nb:STO (103) peaks of the superlattice films with different epitaxial strains. **e** Statistical results of OP epitaxial strains and full-width half maximum (FWHM) with different film thicknesses.

in determining HIF, without changing the lattice mismatch, can effectively preclude serious interference from the changes of the octahedral distortion pattern. Our research exhibits the following highlights: (i) synthesizing double-perovskite La$_2$NiMnO$_6$/La$_2$CoMnO$_6$ superlattices with high cationic oxidation states and $B$-site ordering by ozone-assisted depositing and annealing processes; (ii) achieving a large strain critical thickness (> 90 nm), greatly increasing the strain propagation distances for thin films only with a small lattice mismatch (< 1%); (iii) confirming and regulating the room-temperature HIF in LNMO/LCMO double-perovskite superlattice films by the strain-driven oxygen octahedral distortions in experiments; (iv) revealing the constitutive correlations between octahedral distortions, epitaxial strain, and hybrid improper ferroelectricity. These findings and results complement the experimental cases of perovskite superlattices in HIF studies and demonstrate a promising strategy of strain-driven oxygen octahedral distortions for inducing hybrid improper ferroelectricity in double-perovskite superlattices.

## Results

### The transferring of epitaxial strain in superlattice films

High-quality [(La$_2$NiMnO$_6$)/(La$_2$CoMnO$_6$)]$_n$ (LNMO/LCMO) superlattice films with different thicknesses (SL$_{10, 30, 60, 90, 120}$ nm) were grown on (001)-oriented Nb-doped SrTiO$_3$ (Nb:STO) single-crystal substrate using pulsed laser deposition (PLD). The film growth is monitored by reflection high-energy electron diffraction (RHEED), as shown in Fig. 1a. The ozone-assisted growth method prominently improves the degree of lattice ordering of double perovskites. The thin streaks in RHEED patterns and the pronounced oscillations in RHEED intensity indicate a highly ordered layer-by-layer growth mode during the deposition, which provides good preconditions for epitaxial strain induced by lattice mismatch in the superlattices. The half-order diffraction streaks marked by yellow arrows indicate the twofold superstructure streaks from the double-perovskite structure[34]. As shown in the SL$_{120}$ RHEED patterns with spot-mixed streaks, the epitaxial growth gradually transforms from 2D into 3D mode with the increase of film thickness. The film thickness is precisely controlled by the number of oscillations in RHEED intensity, i.e., one fully crystallized double-perovskite unit cell corresponds to two oscillations. More details of RHEED and the film surface with step terraces are shown in Supplementary Fig. S1.

The thickness and epitaxial structures of superlattice films are determined by high-resolution transmission electron microscopy (HR-TEM). Figure 1b shows the cross-sectional TEM images of 90 nm thick films (SL$_{90}$). The neat registries without lattice diffusion at the interface between the films and substrates confirm the cube-on-cube coherent epitaxy, which is beneficial for propagating epitaxial strain caused by lattice mismatch. According to the extra spots highlighted by the white arrows in FFT patterns, three types of double-perovskite growth modes on Nb:STO substrates (type-I, -II and -III) are determined as [001](001)$_{SL}$//[001](001)$_{Sub}$, [001](110)$_{SL}$//[010](001)$_{Sub}$ and [001](110)$_{SL}$//[100](001)$_{Sub}$, respectively. This special epitaxial growth with mixed crystalline orientations is only usual in LNMO films and prefers generating homogeneous structures in films rather than dislocations due to the unique lattice parameters ($2^{1/2}a \approx 2^{1/2}b \approx c$)[35]. The same extra diffraction spots in SAED are systematically analyzed by a combination of the extinction law and the ordered close-packed structures (Supplementary Figs. S2 and S3). More importantly, the subtle differences in lattice mismatch between the different growth modes will help to maintain the epitaxial strain in the films (Supplementary Fig. S4). Previous studies have demonstrated that the phase stability and microstructure of the LNMO films can be tuned by epitaxial strain[35]. Therefore, perovskite oxide films with multiple growth modes are promising for ferroelectric excitation because the difference within oxygen octahedral distortions can be effectively enhanced by tuning epitaxial strain.

The crystalline structure and macroscopic epitaxial strain of the superlattice films were further explored by high-resolution X-ray diffraction (XRD) characterizations. Figure 1c exhibits the XRD $\theta$−$2\theta$ scans of superlattice films with different thicknesses. The macroscopic out-of-plane (OP) epitaxial strain of thin films is usually determined by this diffraction collected perpendicular to the film surface. All diffraction peaks from superlattice films are detected only along the right of (00$l$)$_{Nb:STO}$ peaks and without any other reflections from spurious phases or randomly oriented grains. This result is consistent with the homogeneous structures from TEM images in despite of multiple growth modes. The diffraction peak around (002)$_{Nb:STO}$ is indexed as (220)/(004) due to the almost identical interplanar distance of $d_{(220)}$ and $d_{(004)}$ for LNMO/LCMO. For simplicity, we adopt (002)$_{pc}$ instead while indexing, according to the pseudocubic (pc) notation[36]. The OP lattice constant $c_{pc}$ of the SL$_{10}$ is calculated as 3.848 Å, which corresponds to the OP compressive strain of − 0.88% compared to the average bulk value of LNMO and LCMO[37]. It should be noted that the strain relaxation process in superlattice films is effectively controlled by film thicknesses, as shown by the clear left shift of (002)$_{pc}$ peaks in the enlarged inset of XRD. Correspondingly, the superlattice constant $c_{pc}$ increases with film thickness, and eventually almost returns to the bulk value as shown by the SL$_{120}$. This epitaxial strain arises from a small lattice mismatch (< 1%) between the films and substrates while showing a large critical thickness (> 90 nm) for the strain transfer, which is quite difficult to achieve in other perovskite oxide films. Therefore, the superlattice technique with multiple growth modes greatly increases the strain propagation distances in thin films and promotes the regulation of epitaxial strain by film thickness, even in a small lattice mismatch system. More importantly, such thickness-dependent epitaxial strain in the films does not change the lattice mismatch, thus the stability of the oxygen octahedral distortion pattern in the sustainable long-range modulation can be greatly ensured.

The reciprocal space mapping (RSM) around the asymmetric reflection (103)$_{pc}$ further determines the macroscopic epitaxial strain and the dependence of strain transfer on film thicknesses. As shown in Fig. 1d, Q$_x$ of the SL$_{10}$ for the direction parallel to the film surface, i.e., in-plane (IP), is aligned with that of the substrates, which indicates the coherent epitaxy of superlattices with full strain. According to the lattice parameters calculated from the Q$_z$ (OP) and Q$_x$ (IP), the superlattice films behave macroscopic IP tensile and OP compressive strain, and the strain relaxation occurs obviously due to the increasing film thickness. The OP compressive strain of the superlattice films is consistent with the results in XRD $\theta$−$2\theta$ scans, and gradually releases with increasing thickness until it almost disappears in the SL$_{120}$. This strain relaxation can also be confirmed by the substantial broadening of the epitaxial peaks with the asymmetry in diffraction intensity. For the IP direction, the relaxation of IP tensile strain occurs only in thick films such as the SL$_{120}$ with a slight horizontal shift. The (103)$_{pc}$ peaks of the superlattice films less than 90 nm are all aligned with the Nb:STO substrates at a constant Q$_x$ of 2.561 nm$^{-1}$, which indicates that the films are fully strained in plane with coherent growth[38]. This phenomenon is related to the stable homogeneous structure with multiple epitaxial modes, resulting in the certain retention of IP strain against the strain relaxation. The XRD phi scans confirm a high degree of IP crystallinity for all superlattice films (Supplementary Fig. S5). However, the OP crystallinity gradually decreases as the film thickness increases, as shown by the trend of FWHM values in Fig. 1e, which implies the reduced epitaxy in thick films. Therefore, the strain relaxation in thick films is closely related to the OP crystallinity since the mismatch strain highly depends on good film epitaxy. Such epitaxial strain stably controlled by the film thickness can provide an effective dimension for modulating the magnitude of oxygen octahedral distortions with fixed rotational modes in perovskite oxides.

Geometric phase analysis (GPA) was performed on TEM images to assess the local strain distribution of the superlattice films at

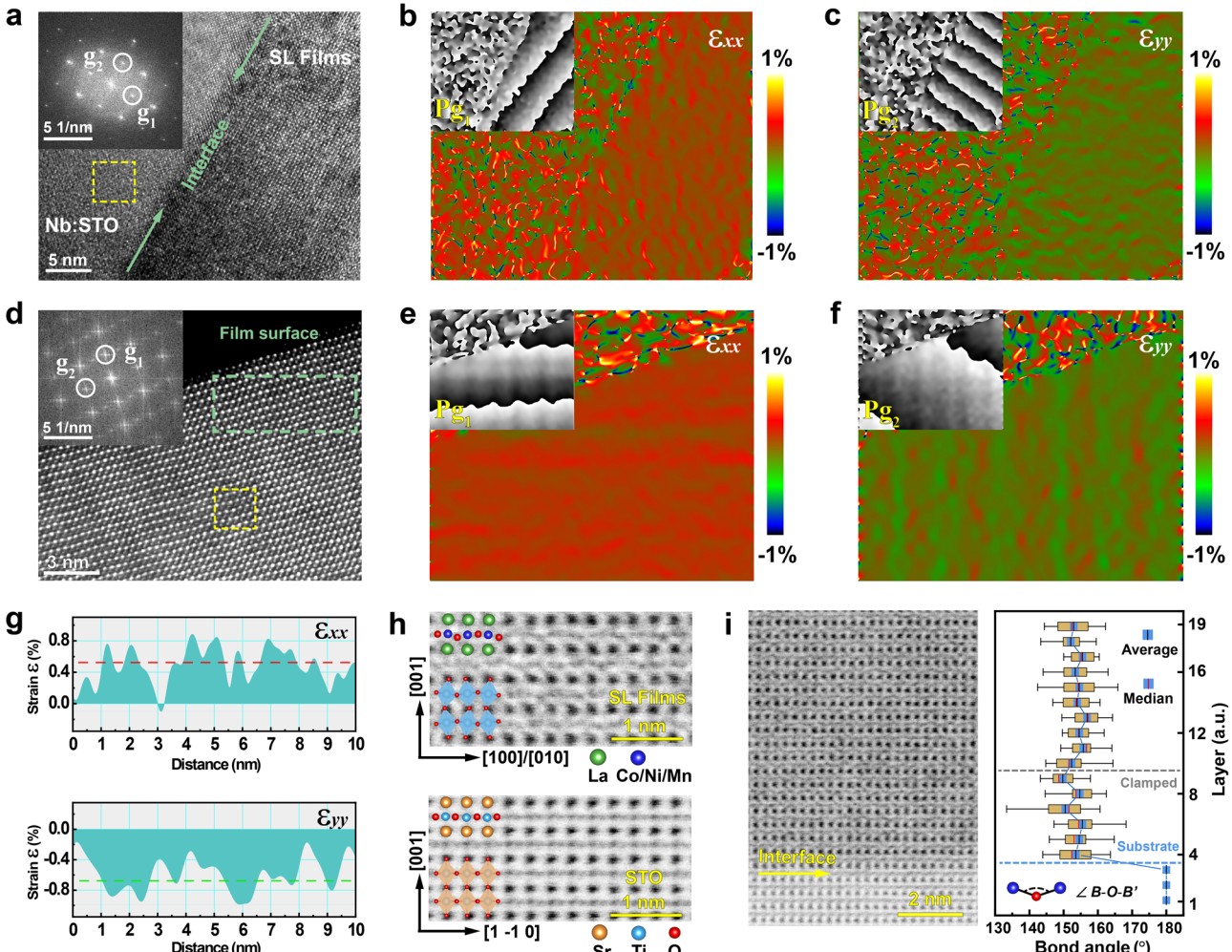

**Fig. 2 | Geometric phase analysis (GPA) and octahedral distortions of the LNMO/LCMO superlattice films. a** Low magnification TEM image of the SL$_{30}$ at the interface and the FFT image of the films. **b, c** Corresponding GPA analysis of (**a**) along in-plane direction $\varepsilon_{xx}$ (**b**) and out-of-plane direction $\varepsilon_{yy}$ (**c**), respectively. **d** HAADF STEM image of the SL$_{60}$ near the surface and the inset of the FFT image. **e, f** Corresponding GPA analysis of $\varepsilon_{xx}$ (**e**) and $\varepsilon_{yy}$ (**f**) on the local atomic image (**d**), respectively. The insets attached to $\varepsilon_{xx}$ and $\varepsilon_{yy}$ are corresponding phase images with normalized phase variation from -π to π (black to white). White circles in FFT images mark the non-collinear reciprocal lattice vectors $\mathbf{g}_1$ and $\mathbf{g}_2$ for GPA. Yellow squares show the reference region for GPA. The color scale indicates the relative difference of local strain in the films. **g** Average intensity profiles of the red and green lines in the grayscale images of $\varepsilon_{xx}$ (**e**) and $\varepsilon_{yy}$ (**f**), respectively. **h** Local annular bright-field (ABF) STEM images of the SL$_{60}$ films and STO. The schematic shows the corresponding $BO_6$ octahedral distortions. **i** The ABF-STEM image of the cross-sectional interface of SL$_{60}$ on STO and the tilting angle ($B$-O-$B'$) of oxygen octahedrons by collecting 19 layers of perovskite unit cells. The average and median are marked by the blue and red lines in the data boxes, respectively. The error bar represents the standard deviation of the tilting angles counted from the ABF image.

nanometer-scale spatial resolution by using FRWR tools. Figure 2a–c show the lattice images and strain maps at the interface between the SL$_{30}$ and substrates. Two non-collinear reciprocal lattice vectors ($\mathbf{g}_1$ and $\mathbf{g}_2$) with good signal-to-noise ratio are chosen in the inset of Fig. 2a for GPA, and the lattices with undistorted fringes are boxed as the reference region to reduce the influence of the different crystal systems between the films and substrates. The clear contrast in strain maps of IP $\varepsilon_{xx}$ and OP $\varepsilon_{yy}$ components (Fig. 2b, c) indicates the uniform IP tensile and OP compressive strains in the films, and the mean square root values of the strain are almost consistent with the result calculated from RSM. As shown in the phase insets of $\varepsilon_{xx}$ and $\varepsilon_{yy}$, the continuous phase gradient in the films indicates the fully coherent growth of superlattices and the epitaxial strain originating from the interface.

To confirm the changes of epitaxial strain with the increasing film thickness, the HAADF-STEM image of the SL$_{60}$ was measured at an atomic-scale resolution near the film surface, as shown in Fig. 2d. The local atomic columns with neat alignment indicate the coherent epitaxy between the superlattice components (Supplementary Fig. S6).

According to the corresponding strain components of $\varepsilon_{xx}$ and $\varepsilon_{yy}$ in Fig. 2e, f, the tensile and compressive strains in the films are along IP and OP orientations, respectively. However, the phase contrasts ($\mathbf{P}\mathbf{g}_1$ and $\mathbf{P}\mathbf{g}_2$) are not perfectly perpendicular to the reciprocal lattice vectors ($\mathbf{g}_1$ and $\mathbf{g}_2$). This is attributed to the diffraction spots with satellite peaks raised from the atomic columns under the multiple epitaxies, reflecting in the macroscopic expression of the phase contrasts. Even so, it should be noted that this phase reflection does not affect the results of the strain analysis because it is mainly carried out in only one phase period (from -π to π) and the atoms in the reference region are well-aligned. The local strain near the film surface is estimated as − 0.7% (OP) and 0.5% (IP), which suggests that the films are fully strained with the propagation of epitaxial strain through the superlattice interlayer interface (Fig. 2g). The intensity profiles of strain maps were processed in grayscale and employed appropriate cutting lines for estimating strain (Supplementary Fig. S7)[39]. The OP strain in the SL$_{60}$ is smaller than that in the SL$_{30}$ when the film thickness increases. This strain relaxation arises from the local misfit dislocation

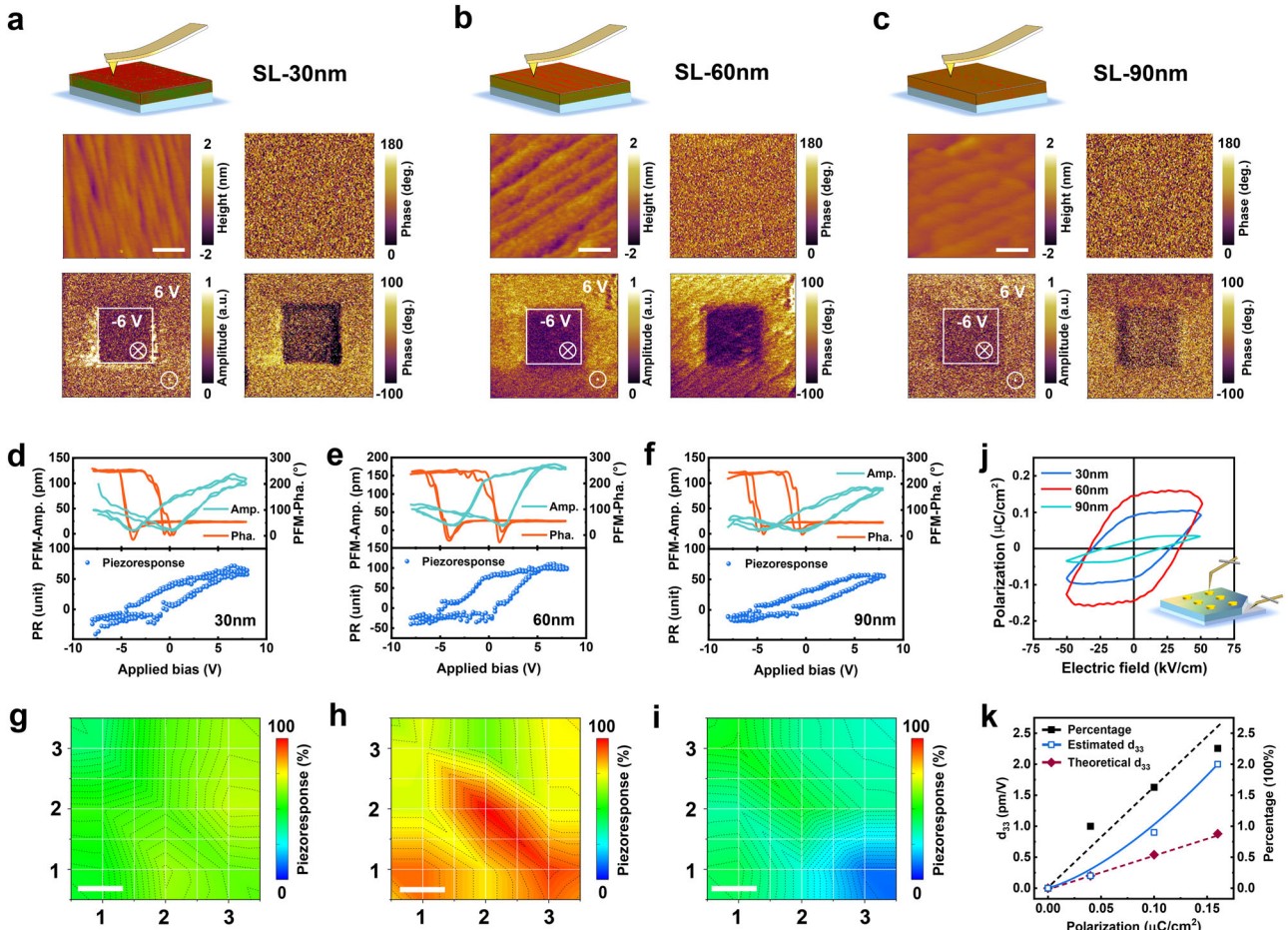

**Fig. 3 | Epitaxial strain-dependent ferroelectric properties and piezoelectric responses of the superlattice films with different thicknesses. a–c** The topography images, initial PFM phase images, and PFM amplitude and phase images after writing domain for the $SL_{30}$ (**a**), $SL_{60}$ (**b**), and $SL_{90}$ (**c**) at the region of $2 \times 2\,\mu m^2$. The box-in-box patterns with ± 6 V are written by a biased conductive tip. The top insets are schematics of the superlattice films with different thicknesses, and the OP and IP strains are distributed according to GPA. The scale bar in topography images for (**a–c**) is 0.5 μm. **d–f** Representative phase–voltage hysteresis loops, amplitude–voltage butterfly loops, and piezoelectric response hysteresis loops of the $SL_{30}$ (**d**), $SL_{60}$ (**e**), and $SL_{90}$ (**f**), respectively. **g–i** Local piezoelectric response distributions of the $SL_{30}$ (**g**), $SL_{60}$ (**h**) and $SL_{90}$ (**i**) at region of $1 \times 1\,\mu m^2$, respectively. The color scale of piezoelectric response (*PR*) is normalized by the value $PR_{max}$ of the $SL_{60}$. The scale bar is 0.25 μm. **j** The polarization-electric field (*P-E*) hysteresis loops of the films with different thicknesses measured by the PUND method. The inset is the positional diagrams of the films and probe during tests. **k** The estimated piezoelectric coefficient $d_{33}$ of the three superlattices based on theoretical approximation and a weight of experimental piezoelectric response.

in the superlattice films due to the reduced epitaxy of thick films (Supplementary Figs. S8 and S9).

Epitaxial strain in $ABO_3$ perovskites can typically induce $BO_6$ octahedral distortions or change the lattice symmetry[40,41]. It is more specific for effecting OOR/OOT in double perovskites with naturally low symmetry. The annular bright-field (ABF) STEM measurement, which is sensitive to light atoms such as oxygen[42], was then performed to visualize $BO_6$ octahedral distortion in the strained $SL_{60}$ films. As shown in Fig. 2h, the oxygen atomic columns of STO in the cross-sectional ABF-STEM image are arranged in straight chains, which is consistent with the 180° bond angle of Ti-O-Ti in *Pm*-3*m* symmetry. While for the LNMO/LCMO superlattices, the elongated oxygen sublattices are arranged in a zigzag-like pattern. This sharp contrast in the oxygen coordination indicates the obvious OOR/OOT in $SL_{60}$ films. Compared to STO substrates with a single crystalline orientation, the partial oxygen sublattices in superlattices are relatively ambiguous due to the phase interfering from multiple growth modes.

Since the apical O atom overlaps with *A*-site La atom, we can only determine the bond angles (*B-O-B'*) associated with OOR by measuring the oxygen positions on the left and right sides of *B*-site atoms. Notably, the ABF image recorded along the STO-[110] axis is the best visualization to analyze OOR (Supplementary Figs. S10–S12). The quantitative bond angles of *B-O-B'* are counted layer by layer within a large region as shown in Fig. 2i. The fluctuation of the bond angle fully reflects the modulation of the octahedral rotation by the epitaxial strain. In addition, the dramatic changes within the six perovskite layers exhibit the clamping effect of the STO substrates on OOR. This limitation of octahedral structures originates from the lattice and symmetry mismatch between the films and substrates, remaining a coherent interface and lattice connectivity[43]. According to the approximately equal values of the median and average of bond angles within a single layer, OOR tends to be stable as the clamping effect disappears. The corresponding bond angle is estimated as ~157° on average, i.e., the octahedron is rotated by approximately 11.5°. Therefore, the epitaxial strains in LNMO/LCMO superlattices have the ability to achieve the sustainable long-range modulation of the oxygen octahedral distortions, especially for OOR/OOT.

## Tunable room-temperature ferroelectric and piezoelectric properties
The room-temperature ferroelectric properties of the superlattice films exhibit good tunability and are characterized by ferroelectric

domain writing/retention and hysteresis loop measurements. Figure 3a–c displays the PFM phase and amplitude images captured from the films with different strains after writing a box-in-box pattern with ± 6 V bias in a $2 \times 2\,\mu m^2$ area. The bright and dark regions in the PFM phase images correspond to the upward and downward polarization states, respectively. All the films with thicknesses from 30 to 90 nm exhibit the opposite phase contrast in writing patterns, indicating the switching behavior of the spontaneous polarization under external electric fields. However, the long-range ferroelectric macrodomains are absent from the initial phase images, replaced by the scattered domains. This result reveals that the room-temperature ferroelectricity of the films is relatively weak and featured with locality. Such scattered domains are even reflected in the writing patterns of the $SL_{30}$ and $SL_{90}$, which is consistent with the previous study on multiferroic materials with improper ferroelectricity[32]. According to the line profiles of the corresponding phase fluctuation, the epitaxial strain has shown strong tunability in ferroelectricity (Supplementary Fig. S13). Generally, it is difficult to induce ferroelectricity in magnetic materials due to the $d^0$ rules, let alone further achieve the regulation of ferroelectricity. More interestingly, the uniform ferroelectric macrodomains and ~180° phase contrast in the $SL_{60}$ show the most robust ferroelectricity under the optimal strain. This result implies that the emergence and regulation of ferroelectricity are primarily caused by strain but with other more intrinsic causations in the mechanism. Subsequently, the stable ferroelectric behavior is further confirmed by the domain retention after three hours with a complex writing pattern (star-in-star) in larger areas of $5 \times 5\,\mu m^2$ (Supplementary Fig. S14). For the $SL_{30}$ and $SL_{90}$, the writing domain in larger areas further indicates the local differences in ferroelectricity (Supplementary Fig. S15), which is related to the unique epitaxial structures of the superlattice films. We cannot obtain the obvious phase contrast from relatively thin or less strained films, such as the $SL_{10}$ and $SL_{120}$ (Supplementary Fig. S16). Therefore, the ferroelectricity of the superlattice films is excited by the strain in a certain thickness region to form stable local domains for testing. Moreover, the same sharp phase contrast of the writing domain on $SL/SrRuO_3/SrTiO_3$ clarifies the nature of such switched behavior originating from the ferroelectric superlattice films (Supplementary Fig. S17).

The local polarization switching characterizations on a single point were further performed to demonstrate the ferroelectricity of the superlattice films and the regulation of epitaxial strain on it. During the poling process, a dc bias of ± 8 V was applied to the Pt-plated tip to reduce the adverse effects of excessive scanning voltage, such as electrostatic interactions and hysteretic surface charging[41]. Figure 3d–f shows the local piezoresponse signals measured on the films with different strains. All films exhibit butterfly-like amplitude loops and square-like phase hysteresis loops, which indicate polarization switching and piezoelectric response characteristics. These opened PFM hysteresis loops change systematically with the epitaxial strain, and the $SL_{60}$ films have stronger ferroelectricity with prominent phase contrast and amplitude saturation. Such robust ferroelectricity corresponds to a large coercive bias, defined as $(V_c^+ - V_c^-)/2$, according to the minimum values of the amplitude loops. This implies that the local ferroelectric domain in the $SL_{60}$ is more difficult to switch due to the high domain switching barrier. Notably, all superlattice films exhibit a clear imprint towards negative bias, defined as $(V_c^+ + V_c^-)/2$, indicating a downward built-in field pointing to the substrates. This similar imprint is generally related to the strain in ferroelectric thin films[44]. Moreover, the absence of PFM hysteresis loops in both $SL_{10}$ and $SL_{120}$ films sufficiently proves the hindrance for ferroelectricity in the relatively thin or less strained films (Supplementary Fig. S18), which is consistent with the results of the writing domain.

On the other hand, the piezoresponse (PR) hysteresis loops of the first-order harmonic displacement can be calculated by the formula of $PR(V) = A(V)\cdot \cos[\varphi(V)]$, where $A(V)$ and $\varphi(V)$ are the amplitude and phase degree, respectively[44], which are acquired at the bias-off state through triangle-square waveform test path (Supplementary Fig. S19). The piezoresponse hysteresis loops represent the piezoelectric response varied with the rotation of electric dipoles[45]. As the same as the analysis of ferroelectricity discussed above, the piezoresponse hysteresis loops in Fig. 3d–f show a strong dependence on the epitaxial strain of the superlattice films, and the maximum value occurs in the $SL_{60}$. To better explore the local ferroelectric response of the superlattice films with different strains, the PFM hysteresis loops are collected from a $3 \times 3$ grid over a $1 \times 1\,\mu m^2$ region of each superlattice film (Supplementary Figs. S20–S23). The corresponding statistical distributions of piezoelectric amplitude, calculated by $(PR^+_{max} - PR^-_{min})/2$, are displayed in Fig. 3g–i. Despite the local fluctuations in piezoresponse, the amplitude distributions overall exhibit an obvious regulation effect from strain, and the $SL_{60}$ shows the maximum amplitude value. Furthermore, the smaller bias imprint of the $SL_{60}$ indicates that the enhanced depolarization field by robust ferroelectricity can weaken the built-in electric field since the surface polarity on Nb-doped substrates guides downward spontaneous polarization (Supplementary Fig. S24). In brief, all the results of PFM measurements confirm the existence of room-temperature ferroelectricity in the superlattice films, and epitaxial strain plays an important role in the excitation and tuning of such ferroelectricity.

It is worth noting that the weak PFM signal in measurements is firmly related to the small remanent polarization, especially in magnetic materials with improper ferroelectricity[33]. To detect the ferroelectric behavior of superlattice films more intuitively, we performed PUND measurements to determine the intrinsic polarization in weak ferroelectric materials since the leakage contributions are enormously mitigated. The P-E hysteresis loops of the three superlattice films in Fig. 3j show the clear ferroelectric saturation feature and significant regulation effect from strain. The $SL_{60}$ exhibits a remnant polarization $P_r$ of ~ 0.16 $\mu C\,cm^{-2}$ measured with a 1 ms pulse width and 1000 ms pulse delays at room temperature. Compared with proper ferroelectrics, such intrinsic weak ferroelectricity is improper in origin, because epitaxial strain usually causes OOR/OOT in double perovskites as confirmed in Fig. 2. Unexpectedly, the value of $P_r$ in this system is in the same order of that in other unconventional ferroelectric $o$-$R$MnO$_3$ films (< 0.5 $\mu C\,cm^{-2}$) with a much lower temperature below 100 K[32], and even close to room-temperature multiferroics γ-$BaFe_2O_4$ (~ 0.2 $\mu C\,cm^{-2}$)[33]. The corresponding integrated current with typical double peaks of polarization switching further demonstrates the intrinsic nature of the ferroelectricity in the films (Supplementary Fig. S25). These results, thus, further indicate the feasibility of regulating room-temperature ferroelectricity in strained magnetic double perovskites. In brief, the thickness-dependent epitaxial strain overcomes the difficulties in tuning epitaxial strain and oxygen octahedral distortion for achieving HIF in the experimental system of perovskite superlattices.

One should emphasize that the coercive fields observed in PUND measurements are quite different from those in PFM hysteresis testing[33]. The former corresponds to equal opposite polarities during the macroscopic polarization reversal under a homogeneous electric field created by the electrodes. While for the later, it corresponds to a piezoelectric response from a still un-switched volume under an inhomogeneous electric field created by the PFM tip. Thus, the probed voltage of domain switching in PFM can be applied substantially higher than that in macroscopic hysteresis measurements. This inference can also be confirmed from the P-E hysteresis loops with large coercive fields detected by the PFM tip (Supplementary Fig. S26). The obvious hysteresis behavior in the $SL_{60}$ proves stronger ferroelectricity than other superlattice films, and the polarization ratio of the superlattice films is similar to the result of PUND measurements. Moreover, the piezoelectric coefficient $d_{33}$ of the films are estimated by combining the dielectric theory approximation with the weighting of

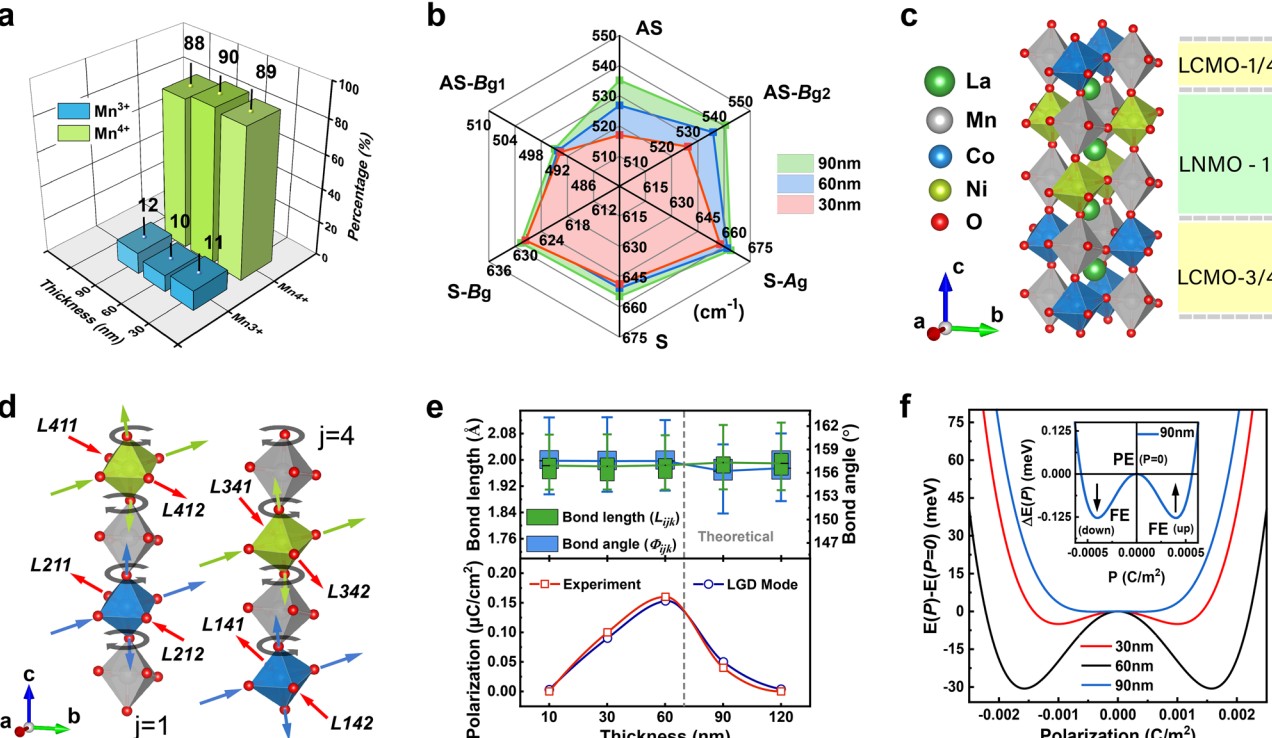

**Fig. 4 | Oxygen octahedral distortion and ferroelectric phase transition of the double-perovskite superlattice films. a** The 3D bar chart of the $Mn^{3+}/Mn^{4+}$ percentage of the $SL_{30}$, $SL_{60}$ and $SL_{90}$ based on XPS core-level spectra of Mn 2p. **b** The radar chart of Raman vibration modes of *B*-O bond including the symmetric stretching (S) and antisymmetric stretching (AS) vibration modes. The spectral peaks of $B_g$ and $A_g$ are from S vibration modes, and the peaks of $B_g^1$ and $B_g^2$ are fitted from AS vibration modes. All number axis represents wavenumber ($cm^{-1}$). **c** The superlattice crystal unit cell with boundary symmetry for DFT simulations. **d** Schematic distortion modes of $BO_6$ octahedron on the representative chains (*j* = 1 and 4) in the double-perovskite unit cell. **e** Statistical results of bond length ($L_{ijk}$), bond angle ($\Phi_{ijk}$), and polarization (*P*) with film thickness for experimental measurement and LGD mode. The gray dashed line marks the theoretical thickness with the most robust ferroelectricity. The error bar represents the standard deviation of the bond length angles counted from the calculated structures. **f** The difference in total free energy of the films as a function of the electric polarization. The inset is a magnification of the energy landscapes of the $SL_{90}$, showing a typical energy barrier of the paraelectric-to-ferroelectric phase transition.

piezoresponse in PFM testing (Supplementary Figs. S27 and S28). As shown in Fig. 3k, $d_{33}$ increases obviously with the polarization in the films, reaching a maximum value of ~ 2.0 pm V$^{-1}$ for $SL_{60}$, which is the same level as the previous study on multiferroics. Therefore, the regulating of the room-temperature ferroelectricity is truly and effectively achieved in the LNMO/LCMO double-perovskite superlattice films. More importantly, these measurement results display a non-monotonic ferroelectric contribution of the strain-regulated octahedral distortion (OOR/OOT), which is a crucial feature for identifying HIF in experimental measurements.

**Strain-driven $BO_6$ octahedral distortion for ferroelectric phase transition**

The degree of *B*-site ordering can severely affect the crystal structures and functional properties of double-perovskite films, especially for LNMO and LCMO. It has been predicted on the theory that structural distortion with *B*-site ordering is beneficial for inversion symmetry breaking (ISB) in double-perovskite bilayers, thereby inducing ferroelectricity[10]. However, the valence state of *B*-site cations will seriously affect *B*-site ordering because the incomplete valence states usually occur near oxygen vacancy and in ion pairs, increasing the anti-site disorder (ASD) in double perovskites[46]. In this work, the optimization of ozone-mixed oxygen pressure processing effectively suppresses the incomplete valence state of *B*-site cations, thereby obtaining high cationic oxidation states and *B*-site ordering. Figure 4a shows a high oxidation state of ~ 90% $Mn^{4+}$ in all superlattice films, according to the XPS core-level spectra and peak fitting of Mn 2p (Supplementary Fig. S29 and Table S1). It is noted that the oxidation

state of Mn can assess the level of the mixed valence states in all transition-metal ions since the high (low) oxidation state corresponds to the ion pairs of $Mn^{4+}/Co^{2+}$ and $Mn^{4+}/Ni^{2+}$ ($Mn^{3+}/Co^{3+}$ and $Mn^{3+}/Ni^{3+}$). Thus, ozone depositing and annealing processes provide excellent oxygen stoichiometry and *B*-site ordering, achieving high-quality thin films for the inducing and regulating of ferroelectricity. Moreover, the disordered distribution of low $Mn^{3+}$ concentration cannot form the charge ordering with $Mn^{4+}$ in superlattice films, which excludes the possibility of charge ordering as a ferroelectric origin in this system.

In addition, such room-temperature ferroelectricity in superlattice films is also not induced by magnetic structures because LCMO or LNMO has transformed into a paramagnetic state at room temperature[16,17]. The displacement-type ferroelectricity usually has a large spontaneous polarization[47], which is inconsistent with the PUND test results. Compared with the unconventional ferroelectricity in manganese oxides, one can suggest that this ferroelectricity is hybrid improper and arises from the tilting or rotation of the oxygen octahedron. As reported, oxygen octahedral distortion is common in perovskite manganese oxides[48], especially in the case of epitaxial strain. Generally, HIF can be theoretically predicted in A/*B*-site ordered double-perovskite superlattice systems, but the experimental difficulties in controlling epitaxial strain and octahedral distortion hinder the inducing of HIF in perovskite superlattices. For the thickness-dependent strain without changing the lattice mismatch, the stability of the octahedral distortion pattern in the sustainable long-range modulation is greatly ensured in LNMO/LCMO superlattices.

The vibration and distortion of the $BO_6$ oxygen octahedra are subsequently detected by using Raman spectroscopy which is a

proven technique for identifying local cation ordering and structural distortions for double-perovskite systems. Figure 4b shows the distribution of Raman spectral peaks obtained from the superlattice films with different strains. Strong modes around 650 and 530 cm⁻¹ are assigned to symmetric stretching (S) and antisymmetric stretching (AS) vibrations of the (Ni/Co/Mn)O₆ octahedra, respectively[49]. Strain has a significant effect on the Raman vibration modes. Both AS and S vibrations shift to lower frequencies with the strain increasing, which corresponds to the phonon softening. Such phonon softening is attributed to the strain-driven oxygen octahedral distortions. According to the fitting of spectral peaks, S vibration is related to Raman-active modes of $(A_g \oplus B_g)$ both correspond to the stretching vibration of oxygen octahedra; and AS vibration is composed of the modes $(B^1_g \oplus B^2_g)$ related to the anti stretching vibration [or Jahn-Teller (JT) stretching mode] of oxygen octahedra and the bending vibration of $B$-O bonds (Supplementary Fig. S30). All films possess a similar narrow linewidth of the S vibration mode (with FWHM ≈ 53 cm⁻¹), indicating the high $B$-site ordering in superlattices. Since there are no other obvious spectral peaks, the Raman-active vibrations are excited by a monoclinic ordered phase. The redshift of AS vibrations dominated by the alteration of $B^2_g$ mode is much more prominent than that of S vibrations, which implies the different responses of Raman vibration modes to various axial strains in double perovskites. $B^2_g$ vibration mode is sensitive to the OP strain, and other modes are sensitive to the IP strain. For the monoclinic ordered phase, S vibration involves both the IP and OP symmetric stretching of $BO_6$ octahedra, while AS vibration involves only IP antisymmetric stretching[50]. Due to the alteration in IP AS modes, the octahedral distortions can be provided with the ability to produce a component of in-phase rotation, which can be combined by another component of JT distortion mode with different inversion centers then leading to the broken inversion symmetry in $B$-site ordered system[48]. Therefore, strain-driven oxygen octahedral distortions are crucial for inducing the hybrid improper ferroelectricity in superlattice films.

We performed DFT calculations to explore the effect of epitaxial strain on the octahedral distortion mode and clarify the function of oxygen octahedral distortion in inducing ferroelectricity. For the sake of simplifying the periodic boundary conditions, the DP superlattice unit cell used for the calculations was built only in the epitaxial mode of type-I, as shown in Fig. 4c. As expected, $BO_6$ octahedra are rotated or tilted within the strained lattice to optimize the alterant La-O distances by reducing the coordination number around La³⁺ cation[51]. Figure 4d shows the representative chains of distorted octahedra. The labeling method of the atomic site ($A_{ijk}$), $B$-O bond length ($L_{ijk}$), and $B$-O-$B'$ bond angle ($\Phi_{ijk}$) is described in Supplementary Fig. S31. MnO₆ octahedra in both LNMO and LCMO layers are featured with JT distortion where the Mn-O bonds are stretched in the ab plane and compressed along the c-axis. For the (Ni/Co)O₆ octahedra, the Ni/Co-O bonds are stretched along the c-axis. However, along the [110] axis, the Ni-O bond is in an antistretched mode with the Co-O bond, which makes the inversion center deviate from the octahedral center. This result is consistent with the analyses of the Raman vibrational modes concerning the symmetry of the in-phase rotation mode. The differences in various octahedral distortions are related to the different electron configuration and coulomb repulsion of $B$-site cations. The coexistence of OOR/OOT and JT distortion is common in perovskites[48], especially for perovskite manganese oxides to stabilize their crystal structure. More importantly, some distortion modes involving OOR and JT distortion can break the inversion symmetry to emerge ferroelectricity in even-layer perovskite systems[48].

Notably, the displacement of O ions is ultimately compensated by the displacement of the $A/B$-site cations, rather than directly inducing ferroelectricity. With the epitaxial strain increasing, the amplitude of octahedral distortions is changed, while the distortion modes are almost remained (Supplementary Fig. S32, and Tables S2 and S3). The

corresponding $B$-O-$B'$ bond angle and $B$-O bond length associated with OOR and OOT as a function of strain are shown by the box plots in Fig. 4e, respectively. Compared to the strain-free state of an oxygen octahedron, the relative amplitude of octahedral rotation is enhanced under compressive strain, while the tilting is suppressed. The average of calculated bond angles for SL₆₀ is similar to the value measured in the ABF-STEM image, indicating the reliability of the calculations. Interestingly, the maximum polarization of the superlattice films occurs around the critical strain corresponding to the sudden changes in the octahedral distortion, indicating the dependence of ferroelectricity on the coupling of OOR and OOT. In Landau free-energy expansion, a trilinear improper coupling among the octahedral rotation, tilting, and polarization could be tuned by epitaxial strains and generate hybrid improper ferroelectricity[7]. Therefore, the most robust ferroelectricity emerges in SL₆₀ with the strongest coupling of OOR and OOT under the optimal strain. Such spontaneous polarization is not the primary order parameter for the ferroelectric-phase transition and is required for the materials with the necessary distortion tendency of $a^- a^- c^+$ in Glazer notation[2,7].

For the bulk of LNMO and LCMO, the ground-state structures of $P2_1/n$ symmetry possess antipolar displacements resulting from the octahedral rotation pattern of $(a^- a^- c^+)$[9]. The enhanced differences in local strain and microstructures can provide the polar mode for the trilinear coupling, so the ferroelectricity in LNMO/LCMO superlattice films is indubitably hybrid improper. Furthermore, combining the lattice distortion and complex epitaxial modes can lead to a symmetry decrease from centrosymmetric $P2_1/n$ to polar $P2_1$ phase through the breaking of the sliding surface $n$. The symmetry changes still maintain the $a^- a^- c^+$ rotation[10]. Therefore, the induced room-temperature improper ferroelectricity in the superlattice films originates from strain-driven oxygen octahedral distortions, and the spontaneous polarization relies on the coupling of nonpolar OOR and OOT modes. To further explore the regulation of epitaxial strain on ferroelectricity, the Landau-Ginzburg-Devonshire (LGD) theory is employed to establish the model of the ferroelectric phase transition for the films (Supplementary Appendix S1). The calculated polarizations of thin films in Fig. 4e are closely consistent with the experimental measurements. The corresponding relationship between the free energy $\Delta E(P)$ and polarization ($P$) under different strains are shown in Fig. 4f. The double well potential reflects the phase transition from ferroelectric (FE) to paraelectric (PE), and the ferroelectric polarization reversal has symmetric energy states. For simplicity, this strain-related ferroelectric mechanism is ideal in a single epitaxial mode. The ferroelectricity in superlattice films is dependent on the local epitaxial structure and strain distribution, which has been indicated by local PFM tests. Thus, due to the coexistence of different domain configurations and local incomplete strain relaxation in thin films, the actual double well potential may be asymmetric[52].

## Discussion

One should be noted that, in theory, HIF is widely accepted to normally occur in the $ABO_3/A'BO_3$ perovskite superlattices rather than $ABO_3/AB'O_3$ systems[4,9,11]. Since $A$-site and $B$-site positions have different inversion symmetry, the octahedral rotations combined with $A/A'$ layered cation ordering can induce effectively polar trilinear term, i.e., the chemical criterion; the rotation pattern of $a^- a^- c^+$ is necessary to dominate the energy landscape over other competing instabilities and drive the transition to the polar structure, i.e., energy criterion[4]. For LNMO/LCMO superlattices with $B$-site cation orderings, it seems to be contrary to the chemical criterion of HIF. However, this double-perovskite superlattice with $a^- a^- c^+$ rotations is much more different from the general systems of $ABO_3/AB'O_3$. Under epitaxial strain, various oxygen octahedrons (MnO₆, CoO₆, and NiO₆) will develop different degrees of rotation/tilting and JT distortion due to the different chemical properties, resulting in the different inversion symmetry of $B$-site

positions. Furthermore, the multimodal epitaxial growth provides a relatively complex lattice environment, driving a polar trilinear term similar to that in the $A/A'$ ordered system. Therefore, in fact, many experimental factors can cause the breaking of inversion symmetry to some extent or locally in perovskite superlattice systems, such as lattice distortions, oxygen vacancies, dislocations, strain, and cation orderings.

Moreover, the $B$-site ordered perovskite superlattice systems have been increasingly focused on the theoretical study of hybrid ferroelectricity and multiferroics. The coexistence of lattice distortions and charge transfer can induce the hybrid ferroelectricity or polar structures in the perovskite superlattices $(ABO_3)_n/(AB'O_3)_n$ ($n = 2$)[53,54]. The combination of the ferroelectricity induced by octahedral rotation and the ferromagnetism/ferrimagnetism caused by the ordered arrangement of different magnetic ions is also calculated in the $B$-site ordered double-perovskite bilayer[10]. In this work, the hybrid improper ferroelectricity in LNMO/LCMO double-perovskite superlattices is induced and regulated by the strain-driven oxygen octahedral distortions, specifically in the Jahn-Teller distortions with coupling octahedral rotation and tilting. For designing HIF in experiments, driving oxygen octahedral rotation/tilting usually requires a large octahedral rotation phase mismatch between the films and substrates. Thus, epitaxial strain plays a very important role in maintaining and enhancing octahedral rotation. The thickness-dependent epitaxial strain maintains a single mode of octahedral distortion during strain regulation, which facilitates the demonstration of non-monotonic ferroelectric contributions, as a crucial feature for identifying HIF in experimental measurements, from the strain-modulated coupling of octahedral rotation and tilting. It seems that the double-perovskite superlattice system of $(A_2BB'O_6)_n/(A_2BB''O_6)_n$ ($n = 1$) could be another platform for achieving HIF through the unique structural design.

In summary, by a strain-driven oxygen octahedral distortion strategy, we report an experimental demonstration of a tunable hybrid improper ferroelectricity (HIF) at room temperature in La$_2$NiMnO$_6$/La$_2$CoMnO$_6$ double-perovskite superlattices. In the optimized films with ~60 nm thickness and 0.72% out-of-plane compressive strain, the remnant polarization ($P_r$) and piezoelectric coefficient ($d_{33}$) are ~0.16 μC cm$^{-2}$ and ~2.0 pm V$^{-1}$, respectively, which compare to the current excellent magnetic unconventional ferroelectrics. The epitaxial growth mode with mixed crystalline orientations relieves the release of epitaxial strain and achieves a large strain critical thickness (> 90 nm) even within a small lattice mismatch (< 1%). Such thickness-dependent strain overcomes the universal difficulties of the sustainable long-range modulation for octahedral distortion in designing HIF under a single octahedral distortion pattern. A hybrid improper mechanism coupling octahedral rotation/tilting and Jahn-Teller distortions is determined. The constitutive correlations between HIF, octahedral distortions, and strain are revealed by a ferroelectric phase transition model based on the Landau-Ginsburg-Devonshire theory. This study confirms the effectiveness of the strain-driven oxygen octahedral distortion strategy for inducing and regulating the hybrid improper ferroelectricity in double-perovskite superlattices and provides an experimental platform and a reliable strategy for overcoming the incompatibility of electronic mechanism in multiferroics.

## Methods

### Epitaxial film growth
The double-perovskite superlattice films [(La$_2$NiMnO$_6$)/(La$_2$CoMnO$_6$)]$_n$ were grown on Nb:SrTiO$_3$ (001) substrates by pulsed laser deposition (PLD) with in-situ reflection high-energy electron diffraction (RHEED) and self-built vacuum transmission system (Loadlock). The background pressure is $6 \times 10^{-8}$ Torr at $T_{substrate} \approx 800$ °C. The pulse laser with a repetition rate of 2 Hz and an energy density of ~2 J/cm$^2$ was emitted by a KrF excimer laser ($\lambda = 248$ nm). The growth temperature and the oxygen partial pressure were 780 °C and 500 mTorr, respectively. A postdeposition ozone annealing process was performed in situ at an oxygen partial pressure of 800 mTorr for 2 h and then cooled to room temperature at a rate of 2 °C min$^{-1}$ to achieve excellent oxygen stoichiometry. The ozone content is regulated by the working power of an ozone generator. The substrates were etched by buffered-HF (pH ≈ 5.0) to get the TiO$_2$-terminated surface.

### Crystal structure and composition characterization
The X-ray diffraction of $\theta$-2$\theta$ scan, rocking curve, ϕ-scan, and reciprocal-space mapping (RSM) was performed by a high-resolution four-circle X-ray diffractometer (X'pert MRD, PANalytical, $\lambda \approx 0.154$ nm). High-resolution lattice images, SAED, and EDS mappings were measured by a field emission transmission electron microscopy (TEM, JEOL, JEM-2100F). High-angle annular dark-field (HAADF) and annular bright-field (ABF) STEM measurements were carried out using a CEOS Cs-corrector operated at 200 kV (Thermo Scientific, Themis Z). The cross-sectional samples for TEM testing were prepared by Gatan PIPS II. The fast Fourier transform (FFT) and geometrical phase analysis (GPA) were performed on DigitalMicrograph software (Gatan, GMS). The chemical composition was determined by a Thermo Scientific K-Alpha+ XPS system with an X-ray emission source of Al Kα (1486.6 eV). Raman spectra signals were recorded in the backscattering configuration using a confocal spectrometer (DXR2Xi, Thermo Fisher Scientific, 532 nm). The size of the confocal pinhole is 25 μm, and the final signals were calibrated by a Si band peak of 520.7 cm$^{-1}$.

### Ferroelectricity and domain structures measurement
The ferroelectric domain structure was confirmed using scanning probe microscopy (Cypher, Asylum Research) with dual-frequency resonant tracking mode. The direct-current bias of ± 6 V was applied to the conductive probes (PPP-EFM, Nanosensors) for switching the ferroelectric domain. The piezoelectric responses were probed by switching spectroscopy PFM (SS-PFM) under the testing bias of ± 8 V. Precision Premier II Ferroelectric Tester (Radiant Technologies) was used to measure P−E hysteresis loops through Positive-Up-Negative-Down (PUND) method, and Au electrodes were plated on the films for measurement. To probe ferroelectricity in microscopic regions, a direct-contact probe station with a micromanipulation conductive probe (MikroMasch, HQ:NSC18/Pt) was designed.

### Density functional theory (DFT) calculation
The plane wave pseudopotential code CASTEP was used for DFT calculations. The Perdew-Burke-Ernzerhof (PBE) of generalized gradient approximation (GGA) and the k-point mesh of $4 \times 4 \times 3$ were chosen for geometric optimization. The plane-wave energy cutoff was 480 eV. A crystal cell of LNMO/LCMO double-perovskite superlattices with boundary symmetry conditions was built for DFT calculation. The applied IP and OP strains were equivalently replaced by the changed lattice parameters in the experiment. Based on the optimization results of the strained crystal structures, the oxygen positions can be determined, and the $BO_6$ octahedral deformation and OOR/OOT are further determined by measuring the $B$-O bond lengths and $B$-O-$B$ bond angles, respectively.

### Reporting summary
Further information on research design is available in the Nature Portfolio Reporting Summary linked to this article.

## Data availability
The authors declare that all data supporting the findings of this study are available within the paper and its supplementary information files.

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

## Acknowledgements

This work was financially supported by the National Natural Science Foundation of China (Grant Nos. 12074204, 12374258 and 62205011), Natural Science Foundation of Inner Mongolia (Grant No. 351 2022ZD06), Program for Young Talents of Science and Technology in Universities of Inner Mongolia Autonomous Region (No. NMGIRT2203), and Fundamental Research Funds for the Central Universities (No. buctrc202122), the Singapore National Research Foundation (No. NRF-CRP22-2019-0004).

## Author contributions

Conceptualization: Y.X.J. and S.F.Z. Methodology: Y.X.J., J.G.N., C.W., D.L.X., and X.H.S. Investigation: Y.X.J. and J.G.N. Writing—original draft: Y.X.J. Writing—review & editing: C.W., W.B.G., and S.F.Z. Funding acquisition: C.W., W.B.G., and S.F.Z. All authors discussed the results and contributed to the paper. All authors read and approved the final manuscript.

## Competing interests

The authors declare no competing interests.
