## [Peer Review File · Nature Communications]

Experimental demonstration of tunable hybrid improper ferroelectricity in double-perovskite superlattice filmsREVIEWER COMMENTS

Reviewer #1 (Remarks to the Author):

1. Abstract: The authors should write a more comprehensive abstract with the clear-cut idea of their work and output in a quantitative form. In addition, More sentences should be used to describe the results.
2. Introduction: In the literature review, authors should briefly present some of the existing relevant work emphasizing on the gap between the current work and the existing literature and also should demonstrate how the current work will fill this gap.
3. The novelty of this work was not specified, authors should be discussed about the novelty of their work.
4. Please highlight/indicate the objectives and limitations of this study in detail at the last paragraph of the Introduction.
5. Conclusion: Presentation of the conclusion is not good. The author should write the output in a quantitative form.

Reviewer #2 (Remarks to the Author):

The found and proof of the room-temperature hybrid improper ferroelectricity in La₂NiMnO₆/La₂CoMnO₆ films is interesting. The authors fabricated high-quality double-perovskite films and characterized the ferroelectricity and piezoresponse of the films successfully. However, the strategy of using different layers of superlattice structure to induce the strain lacks of the innovation. And there are some concerns which need to be addressed before the publication in this high-quality journal:

- 1) The evolution of the BO₆ octahedron tilting/rotation could reflect through the SAED (which is absence in the manuscript or supporting information) or the FFT images (which shows no evidence of the BO₆ octahedron in the Fig. 1b). It is possibly that the concepts of SAED and FFT are confused in the article.
- 2) For the ϵ_{xx} used the vector g_1 which is perpendicular to the interface, is it possible the Fig. 2b indicates the situation of out-of-plane strain? Or there may be some mistakes in the annotation of the images. Meanwhile, the stripe-like contrast in the grayscale of Fig. 2c should also be perpendicular to its reciprocal lattice vectors. Thus, all the GPA analysis need to be checked carefully.
- 3) The conclusion of the tilting/rotation of BO₆ octahedron in the superlattice films made by Fig. 2h need to be reconsidered, at least providing the quantitative data. The discrepancy of the displacement is difficult to observe only by the individual atomic-scale HAADF-STEM image.

Below is a list of the main changes to the manuscript:

- (1) Introduction: The existing relevant works are presented and summarized to emphasize the gap on HIF study to highlight the novelty of our work and how to fill the gap. The objectives and limitations of this work are indicated in detail at the last paragraph.
- (2) Abstract: The related quantitative results and necessary descriptions are added, in response to Reviewer #1's comments.
- (3) Main text: There are several important changes in the text to comply with the comments raised by the reviewers, including innovation highlights, mechanism analysis and experimental supports, which have been discussed in the point-to-point response. Conclusion is revised as the output in a quantitative form according to the Reviewer #1's comments.
- (4) Figures and captions: we update the calculation results on GPA and add the ABF-STEM measurements in Fig. 2 to accurately visualize the strain and quantitatively discuss the oxygen octahedral rotation (OOR).
- (5) Supporting online materials: we add the results of SAED and EDS mappings in Supplementary Figures S2 and S3, respectively, in order to double-perovskite superstructures. The local HAADF image and strain visualizations are added in Supplementary Figures S6, S7 and S9. The details and discussions of ABF-STEM measurements are displayed in Supplementary Figures S10 to S12.
- (6) References: we add six highly relevant references to improve our manuscript on the comparison of ferroelectric properties and quantitative analysis of OOR.
- (7) Methods: The details of film growth, SAED patterns, EDS mappings and ABF-STEM measurements are added.

In the following, the reviewer's original comments are shown by black italic characters. Our Responses are shown by blue characters, and Revisions are shown by red characters.

Response to Reviewers' Comments

We appreciate the reviewers' time and effort for reviewing our manuscript. The reviewers' comments and suggestions were very constructive and helped us to improve our manuscript. The following is point-to-point response to their comments. We have properly addressed all the questions from the reviewers and made necessary revisions in the revised manuscript. All the revisions in the revised manuscript are highlighted.

Reviewer #1

Thanks for the reviewer's comprehensive review and constructive comments. We fully agree with the reviewer that the necessary statements of innovations and relevant discussions, as well as quantitative presentations of experimental results should be added or highlighted in our manuscript, including Abstract, Introduction, Discussion and Conclusion. According to reviewer's comments, we have responded to the related questions and revised the manuscript. The corresponding point-to-point responses to reviewer's comments are as follows:

Comment 1: *Abstract: The authors should write a more comprehensive abstract with the clear-cut idea of their work and output in a quantitative form. In addition, more sentences should be used to describe the results.*

Response: We thank the reviewer for this valuable suggestion. We fully agree with the reviewer that a quantitative form and more descriptions for the results can better exhibit the content and research ideas of this work. According to the comments, we have added the necessary quantitative results and related descriptions in the Abstract.

Revision: The new sentences "The epitaxial growth mode with mixed crystalline orientations maintains a large strain transfer distance more than 90 nm in the superlattice films with lattice mismatch less than 1%. Such epitaxial strain permits sustainable long-range modulation of oxygen octahedral rotation and tilting, thereby inducing and regulating HIF. A robust room-temperature HIF with remnant polarization of $\sim 0.16 \mu\text{C cm}^{-2}$ and piezoelectric coefficient of 2.0 pm V^{-1} is obtained,

and the DFT calculations and Landau-Ginsburg-Devonshire theory reveal the constitutive correlations between HIF, octahedral distortions, and strain.” were added in the Abstract (page 2/lines 8-15).

***Comment 2:** Introduction: In the literature review, authors should briefly present some of the existing relevant work emphasizing on the gap between the current work and the existing literature and also should demonstrate how the current work will fill this gap.*

Response: Thanks for the reviewer’s constructive comments. We fully agree with these viewpoints. A brief presentation of relevant previous work is necessary to emphasize the gap in HIF research and highlight the importance and innovation of our research contents. The necessary details of the brief presentation and analysis on the existing literature are as follows:

So far, studies of hybrid improper ferroelectricity (HIF) in perovskite superlattices are still stagnant at the theoretical level⁹⁻¹¹, and there is still no experimental case reported for perovskite superlattice systems with well-defined HIF. We have summarized two main factors on experiment that hinder the design of HIF in perovskite superlattices. On the one hand, the fabrication of high-quality double-perovskite superlattices is challenging but necessary for identifying HIF^{18, 19}. On the other hand, strain-induced structural distortions are quite complex and difficult in tailoring²⁴, and the epitaxial strain in thin films not only depends on the excitation of large lattice mismatch^{25, 26}, but also decays rapidly with increasing film thickness^{27, 28}. In our revised manuscript, the above existing relevant work have been briefly presented to emphasize the research gap on HIF in the Introduction.

For the first factor, by using ozone-assisted depositing and annealing processes, we synthesized double-perovskite $\text{La}_2\text{NiMnO}_6/\text{La}_2\text{CoMnO}_6$ superlattice films with high oxidation states and ordering of *B*-site cations. For the second factor, the epitaxial growth mode with mixed crystalline orientations slows down the rapid release of epitaxial strain with film thickness and increases the strain critical thickness as large as 90 nm. Meanwhile, film thickness-dependent epitaxial strain ensures the stability of the oxygen octahedral distortion pattern in the sustainable long-range modulation.

Such strain regulation effectively precludes serious interference in the determination of HIF by changes in octahedral distortion pattern raised from various lattice mismatches commonly for changing the epitaxial strain. The relevant demonstration of filling the current gap has been added before the last paragraph of the Introduction.

In brief, we overcome the difficulties in tuning the epitaxial strain and oxygen octahedral distortion for achieving HIF in experimental system of perovskite superlattices, i.e., a robust room-temperature HIF with remnant polarization of $\sim 0.16 \mu\text{C cm}^{-2}$ and piezoelectric coefficient of $\sim 2.0 \text{ pm V}^{-1}$ in $\text{La}_2\text{NiMnO}_6/\text{La}_2\text{CoMnO}_6$ superlattice films. This work addresses the gap in HIF experimental studies of perovskite superlattices and demonstrates a promising strategy of strain-driven oxygen octahedral distortions for inducing HIF in double-perovskite superlattices.

Thanks again for the reviewer's valuable suggestion which highlights the research content and innovation of our work.

References:

9. H. J. Zhao, et al. Near room-temperature multiferroic materials with tunable ferromagnetic and electrical properties. *Nat. Commun.* **5**, 4021 (2014).
10. J. Zhang, et al. Design of Two-dimensional multiferroics with direct polarization-magnetization coupling. *Phys. Rev. Lett.* **125**, 017601 (2020).
11. X. Shen, F. Wang, X. Lu & J. Zhang. Two-dimensional multiferroics with intrinsic magnetoelectric coupling in *A*-site ordered perovskite monolayers. *Nano Lett.* **23**, 735-741 (2022).
18. M. P. Singh, K. D. Truong, S. Jandl & P. Fournier. Long-range Ni/Mn structural order in epitaxial double perovskite $\text{La}_2\text{MnNiO}_6$ thin films. *Phys. Rev. B* **79**, 224421 (2009).
19. M. Kitamura, et al. Ferromagnetic properties of epitaxial $\text{La}_2\text{MnNiO}_6$ thin films grown by pulsed laser deposition. *Appl. Phys. Lett.* **94**, 132506 (2009).
24. W. Li, et al. Atomic-scale control of electronic structure and ferromagnetic insulating state in perovskite oxide superlattices by long-range tuning of BO_6 octahedra. *Adv. Funct. Mater.* **30**, 2001984 (2020).
25. S. Salmani-Rezaie, K. Ahadi, W. M. Strickland & S. Stemmer. Order-disorder ferroelectric transition of strained SrTiO_3 . *Phys. Rev. Lett.* **125**, 087601 (2020).
26. W. Peng, et al. Constructing polymorphic nanodomains in BaTiO_3 films via epitaxial symmetry engineering. *Adv. Funct. Mater.* **30**, 1910569 (2020).
27. Z. Liao, et al. Controlled lateral anisotropy in correlated manganite heterostructures by interface-engineered oxygen octahedral coupling. *Nat. Mater.*

15, 425-431 (2016).

28. Z. Zhang, et al. Uniaxial strain and hydrostatic pressure engineering of the hidden magnetism in $\text{La}_{1-x}\text{Ca}_x\text{MnO}_3$ ($0 \leq x \leq 1/2$) thin films. *Nano Lett.* **22**, 7328-7335 (2022).

Revision:

The revised sentences “In general, there are two main factors on experiment that hinder the design of HIF in double-perovskite superlattices. Firstly, the growth window of B-site ordered double perovskites is quite narrow^{18, 19}, and during the film growth, the incomplete oxidation state of transition metal cations inevitably leads to various lattice defects²⁰⁻²³.” were added in page 4/lines 9-12.

The revised sentence “Secondly, strain-induced structural distortions are quite complex and difficult in tailoring²⁴; and the epitaxial strain in thin films not only depends on the large lattice mismatch^{25, 26}, but also decays rapidly with increasing film thickness^{27, 28}.” was added in page 4/lines 14-17.

The new sentence “Notably, for determining HIF, the thickness-dependent epitaxial strain can maintain a single mode of octahedral distortion during strain regulation, which facilitates the demonstration of non-monotonic ferroelectric contributions from the coupling of OOR and OOT.” was added in page 4/lines 24-27.

Comment 3: *The novelty of this work was not specified, authors should be discussed about the novelty of their work.*

Response: Thanks for the reviewer’s comments. According to the reviewer’s advice, we have highlighted the innovation discussion in our revised manuscript. The discussion details of the comprehensive innovation analysis are as follows:

In theoretical calculations, hybrid improper ferroelectricity (HIF) has been predicted in several double-perovskite superlattice systems^{10, 11}. However, it is quite difficult to fabricate high-quality double-perovskite superlattices in experiments. Even worse, the strain-induced structural distortion is difficult in tailoring²⁴, and the non-continuous epitaxial strain derived from large lattice mismatch decays rapidly with the film thickness^{25, 28}. Therefore, the difficulties in stability of the sustainable long-range

modulation on octahedral distortion hinder the induction and exploration of HIF in perovskite superlattices. The relevant statements have been also revised and added in the Introduction.

In our work, the above common difficulties in experiment for studying HIF in perovskite superlattices are effectively overcome. We synthesized double-perovskite $\text{La}_2\text{NiMnO}_6/\text{La}_2\text{CoMnO}_6$ superlattice films with high oxidation states and ordering of *B*-site cations by ozone-assisted depositing and annealing processes, which is illustrated in the structure and XPS analysis. By the epitaxial growth mode with mixed crystalline orientations, a large strain critical thickness (>90 nm) is achieved to permit the sustainable long-range modulation of oxygen octahedral rotation/tilting (OOR/OOT). Therefore, the thickness-dependent epitaxial strain maintains the same mode of octahedral distortion during strain modulation, which facilitates the demonstration of non-monotonic ferroelectric contributions from the strain-regulated coupling of octahedral rotation and tilting. It is a crucial feature for identifying HIF in experimental measurements.

These are the important novelty of the long-range modulation of OOR without changing octahedral distortion mode. According to the reviewer's advice, we have specified the novelty in the revised manuscript. The relevant statements on the novelty of the sustainable long-range modulation for octahedral distortion had been discussed and added in the X-ray diffraction analysis, Introduction, and Conclusion.

Furthermore, we have definitively confirmed the HIF and regulated it in LNMO/LCMO superlattices through experiments and structural optimization. The obtained robust room-temperature HIF with remnant polarization of $\sim 0.16 \mu\text{C cm}^{-2}$ and piezoelectric coefficient of 2.0 pm V^{-1} is comparable to that in the current excellent magnetic unconventional ferroelectrics^{32, 33}. These excellent properties are emphasized and discussed once again in PFM and PUND measurement. More importantly, a hybrid improper mechanism with coupling octahedral rotation/tilting and Jahn-Teller distortions is determined, according to experimental measurements and DFT calculations. Using Landau-Ginsburg-Devonshire theory, a ferroelectric phase transition model is established to reveal the constitutive correlations between

HIF, octahedral distortions, and strain. This work demonstrates a reliable strategy of strain-driven oxygen octahedral distortions for inducing HIF in double-perovskite superlattices and provides a promising experimental platform for overcoming the incompatibility of electronic mechanism in multiferroics.

These are the important novelty of revealing a hybrid improper mechanism on tunable HIF at room temperature. According to the reviewer's advice, we have specified the novelty in the revised manuscript. The corresponding emphasis and discussion were specified and added in the DFT calculations, ferroelectric phase transition analysis, and Conclusion.

Thanks again for the reviewer's suggestion. The related revisions have greatly highlighted the innovation of our work and made the manuscript significantly more readable.

References:

10. J. Zhang, et al. Design of Two-dimensional multiferroics with direct polarization-magnetization coupling. *Phys. Rev. Lett.* **125**, 017601 (2020).
11. X. Shen, F. Wang, X. Lu & J. Zhang. Two-dimensional multiferroics with intrinsic magnetoelectric coupling in *A*-site ordered perovskite monolayers. *Nano Lett.* **23**, 735-741 (2022).
24. W. Li, et al. Atomic-scale control of electronic structure and ferromagnetic insulating state in perovskite oxide superlattices by long-range tuning of BO_6 octahedra. *Adv. Funct. Mater.* **30**, 2001984 (2020).
25. S. Salmani-Rezaie, K. Ahadi, W. M. Strickland & S. Stemmer. Order-disorder ferroelectric transition of strained $SrTiO_3$. *Phys. Rev. Lett.* **125**, 087601 (2020).
28. Z. Zhang, et al. Uniaxial strain and hydrostatic pressure engineering of the hidden magnetism in $La_{1-x}Ca_xMnO_3$ ($0 \leq x \leq 1/2$) thin films. *Nano Lett.* **22**, 7328-7335 (2022).
32. E. M. Choi, et al. Nanoengineering room temperature ferroelectricity into orthorhombic $SmMnO_3$ films. *Nat. Commun.* **11**, 2207 (2020).
33. F. Orlandi, et al. γ - $BaFe_2O_4$: a fresh playground for room temperature multiferroicity. *Nat. Commun.* **13**, 7968 (2022).

Revision:

The new sentence “More importantly, such thickness-dependent epitaxial strain in the films does not change the lattice mismatch, thus the stability of the oxygen octahedral distortion pattern in the sustainable long-range modulation can be greatly ensured.”

was added in page 7/lines 24-27.

The new sentence “In brief, the thickness-dependent epitaxial strain overcomes the difficulties in tuning epitaxial strain and oxygen octahedral distortion for achieving HIF in experimental system of perovskite superlattices.” was added in page 14/lines 1-3.

The new sentences “Generally, HIF can be theoretically predicted in A/B-site ordered double-perovskite superlattice systems, but the experimental difficulties in controlling epitaxial strain and octahedral distortion hinder the inducing of HIF in perovskite superlattices. For the thickness-dependent strain without changing the lattice mismatch, the stability of octahedral distortion pattern in the sustainable long-range modulation is greatly ensured in LNMO/LCMO superlattices.” were added in page 15/lines 25-30.

Comment 4: *Please highlight/indicate the objectives and limitations of this study in detail at the last paragraph of the Introduction.*

Response: Thanks for the reviewer’s reasonable comments. We agree with the reviewer that it is necessary to highlight/indicate the objectives and limitations of this study in detail. According to the reviewer’s advice, we have indicated the objectives and limitations of our work on HIF in the last paragraph of the Introduction, and the necessary references were added. The details of the related demonstrations are as follows.

In this work, we overcame the universal difficulties of the sustainable long-range modulation for octahedral distortion in designing HIF by regulating thickness-dependent epitaxial strain, and achieved a tunable HIF in LNMO/LCMO double-perovskite superlattices. The obtained room-temperature HIF behaves a remnant polarization of $\sim 0.16 \mu\text{C cm}^{-2}$ and a piezoelectric coefficient of 2.0 pm V^{-1} . Although these properties still have a non-negligible shortfall with conventional ferroelectrics²⁹⁻³¹, it is comparable to the current excellent magnetic unconventional ferroelectrics^{32,33}. This limitation of our work has been indicated in the manuscript.

The objectives and highlights of this work on HIF were summarized as follows: (i) synthesizing double-perovskite $\text{La}_2\text{NiMnO}_6/\text{La}_2\text{CoMnO}_6$ superlattices with high cationic oxidation states and *B*-site ordering by ozone-assisted depositing and annealing processes; (ii) achieving a large strain critical thickness (>90 nm), greatly increasing the strain propagation distances for thin films only with a small lattice mismatch ($<1\%$); (iii) first confirming and regulating the room-temperature HIF in LNMO/LCMO double-perovskite superlattice films by the strain-driven oxygen octahedral distortions in experiments; (iv) revealing the constitutive correlations between octahedral distortions, epitaxial strain, and hybrid improper ferroelectricity.

In brief, although the properties of HIF in LNMO/LCMO superlattices still have a non-negligible shortfall with conventional ferroelectrics, these findings and results first complement the experimental cases of perovskite superlattices in HIF studies. We thank the reviewer again for the valuable suggestion. The added details of the objectives and limitations on our HIF study at the last paragraph of the Introduction have improved our manuscript more comprehensively.

References:

29. M. Fang, et al. Tuning the interfacial spin-orbit coupling with ferroelectricity. *Nat. Commun.* **11**, 2627 (2020).
30. R. Guo, et al. Continuously controllable photoconductance in freestanding BiFeO_3 by the macroscopic flexoelectric effect. *Nat. Commun.* **11**, 2571 (2020).
31. M. F. Sarott, M. D. Rossell, M. Fiebig & M. Trassin. Multilevel polarization switching in ferroelectric thin films. *Nat. Commun.* **13**, 3159 (2022).
32. E. M. Choi, et al. Nanoengineering room temperature ferroelectricity into orthorhombic SmMnO_3 films. *Nat. Commun.* **11**, 2207 (2020).
33. F. Orlandi, et al. $\gamma\text{-BaFe}_2\text{O}_4$: a fresh playground for room temperature multiferroicity. *Nat. Commun.* **13**, 7968 (2022).

Revision:

The new sentences “In this work, we overcame the universal difficulties of the sustainable long-range modulation for octahedral distortion in designing HIF by regulating thickness-dependent epitaxial strain, and achieved a tunable HIF at room

temperature in La₂NiMnO₆/La₂CoMnO₆ double-perovskite superlattices. Although the obtained room-temperature HIF still have a non-negligible shortfall with conventional ferroelectrics²⁹⁻³¹, it is comparable to the current excellent magnetic unconventional ferroelectrics^{32, 33}.” were added in page 4/line 28 to page 5/line 2.

The new sentences “A large strain critical thickness is achieved by the epitaxial growth mode with mixed crystalline orientations, which permits the sustainable long-range modulation of oxygen octahedral rotation and tilting. Such strain-driven octahedral distortion strategy in determining HIF, without changing the lattice mismatch, can effectively preclude serious interference from the changes of octahedral distortion pattern.” were added page 5/lines 4-9.

Added Reference:

29. M. Fang, et al. Tuning the interfacial spin-orbit coupling with ferroelectricity. *Nat. Commun.* **11**, 2627 (2020).
30. R. Guo, et al. Continuously controllable photoconductance in freestanding BiFeO₃ by the macroscopic flexoelectric effect. *Nat. Commun.* **11**, 2571 (2020).
31. M. F. Sarott, M. D. Rossell, M. Fiebig & M. Trassin. Multilevel polarization switching in ferroelectric thin films. *Nat. Commun.* **13**, 3159 (2022).

Comment 5: *Conclusion: Presentation of the conclusion is not good. The author should write the output in a quantitative form.*

Response: Thanks for the reviewer’s comments. We agree with the reviewer’s suggestion on quantitative form of Conclusion, which will make material properties and study innovations much clearer. We have revised the Conclusion in our manuscript, especially for the results and properties output in a quantitative form.

Revision: The new sentences “In summary, by a strain-driven oxygen octahedral distortion strategy, we report experimental demonstration of a tunable hybrid improper ferroelectricity (HIF) at room temperature in La₂NiMnO₆/La₂CoMnO₆ double-perovskite superlattices. In the optimized films with ~ 60 nm thickness and 0.72% out-of-plane compressive strain, the remnant polarization (P_r) and piezoelectric

coefficient (d_{33}) are $\sim 0.16 \mu\text{C cm}^{-2}$ and $\sim 2.0 \text{ pm V}^{-1}$, respectively, which compare to the current excellent magnetic unconventional ferroelectrics. The epitaxial growth mode with mixed crystalline orientations relieves the release of epitaxial strain and achieves a large strain critical thickness ($> 90 \text{ nm}$) even within small lattice mismatch ($< 1\%$). Such thickness-dependent strain overcomes the universal difficulties of the sustainable long-range modulation for octahedral distortion in designing HIF under a single octahedral distortion pattern. A hybrid improper mechanism coupling octahedral rotation/tilting and Jahn-Teller distortions is determined. The constitutive correlations between HIF, octahedral distortions, and strain are revealed by a ferroelectric phase transition model based on the Landau-Ginsburg-Devonshire theory. This study confirms the effectiveness of the strain-driven oxygen octahedral distortion strategy for inducing and regulating the hybrid improper ferroelectricity in double-perovskite superlattices, and provides an experimental platform and a reliable strategy for overcoming the incompatibility of electronic mechanism in multiferroics.” were added as Conclusion (page 19/line 19 to page 20/line 7).

Reviewer #2

Comments: *The found and proof of the room-temperature hybrid improper ferroelectricity in $\text{La}_2\text{NiMnO}_6/\text{La}_2\text{CoMnO}_6$ films is interesting. The authors fabricated high-quality double-perovskite films and characterized the ferroelectricity and piezoresponse of the films successfully. However, the strategy of using different layers of superlattice structure to induce the strain lacks of the innovation. And there are some concerns which need to be addressed before the publication in this high-quality journal.*

Response: Thanks for the reviewer's positive comments, especially for elaborating on the related issues and providing helpful suggestions in the comment report. According to the **Comments 1 to 3**, we have responded to all the questions and revised certain inaccurate expressions accordingly, including the added SAED measurement, the accurate analysis of GPA and the ABF-STEM characterization for quantitative octahedral rotations.

Before the point-to-point responses, we have to apologize for the inadequate statements of the innovations on regulating and determining HIF in the LNMO/LCMO superlattices by changing film thicknesses. It seems that the strategy of only using different layers of superlattice structure to regulate strain is common and lacks of innovation. Indeed, the epitaxial strain regulated by the film thickness is critical for inducing and confirming the HIF in LNMO/LCMO superlattices with specific growth modes.

On the one hand, the thickness-dependent epitaxial strain maintains a single mode of octahedral distortion during strain modulation, which facilitates the demonstration of non-monotonic ferroelectric contributions from the strain-regulated coupling of octahedral rotation and tilting. It is a crucial feature for identifying HIF in experimental measurements. In general, strain-induced structural distortions are quite complex and difficult in tailoring²⁴, and the epitaxial strain in thin films usually depends on the excitation of large lattice mismatch^{25, 26}. Therefore, a great stability of the sustainable long-range modulation on octahedral distortion within a single distortion pattern is the precondition for inducing and identifying HIF in perovskite superlattices. However, replacing/changing the substrates or its orientations will

inevitably alter the epitaxial growth modes and octahedral target rotation pattern, since the phase/lattice mismatch between film materials and substrates is fixed. The non-monotonicity of HIF with strain will not be confirmed for the system with different distortion modes. In this work, the thickness-dependent epitaxial strain does well solve the above problem, since such strain regulation does not change any mismatches of the system, thereby ensuring a single octahedral rotation pattern for identifying HIF. We have added the relevant discussion and analysis in Introduction and the section of HIF test.

On the other hand, this epitaxial strain with a large critical thickness provides the sustainable long-range modulation of octahedral rotation/tilting for regulating HIF. Commonly, the epitaxial strain decays rapidly with the increasing of film thickness^{27, 28}. However, in our work, a epitaxial growth mode with mixed crystalline orientations for double perovskites slows down the rapid release of epitaxial strain with film thickness and greatly increases the strain critical thickness (>90 nm) even in a small lattice mismatch ($< 1\%$). Therefore, the strategy of thickness-dependent epitaxial strain effectively overcomes the difficulties in the long-range regulation of the strain-driven oxygen octahedral rotation/tilting within a single distortion pattern to achieve and confirm HIF. We have added the relevant emphasis and discussion in Introduction, Conclusion and the analysis section of epitaxial strain and HIF.

In brief, the thickness-dependent epitaxial strain solved the difficulties in stabilizing the long-range modulation of octahedral rotation and achieving a large critical thickness of strain transmission, thereby inducing and determining HIF in the LNMO/LCMO superlattices. Thanks for the reviewer's comments again. The corresponding innovation statements on strain-driven octahedral distortion regulated by film thickness for achieving and confirming HIF has been carefully revised and added in our manuscript as shown in the following **Revision**.

References:

24. W. Li, et al. Atomic-scale control of electronic structure and ferromagnetic insulating state in perovskite oxide superlattices by long-range tuning of BO_6 octahedra. *Adv. Funct. Mater.* **30**, 2001984 (2020).
25. S. Salmani-Rezaie, K. Ahadi, W. M. Strickland & S. Stemmer. Order-disorder

- ferroelectric transition of strained SrTiO₃. *Phys. Rev. Lett.* **125**, 087601 (2020).
26. W. Peng, et al. Constructing polymorphic nanodomains in BaTiO₃ films via epitaxial symmetry engineering. *Adv. Funct. Mater.* **30**, 1910569 (2020).
 27. Z. Liao, et al. Controlled lateral anisotropy in correlated manganite heterostructures by interface-engineered oxygen octahedral coupling. *Nat. Mater.* **15**, 425-431 (2016).
 28. Z. Zhang, et al. Uniaxial strain and hydrostatic pressure engineering of the hidden magnetism in La_{1-x}Ca_xMnO₃ ($0 \leq x \leq 1/2$) thin films. *Nano Lett.* **22**, 7328-7335 (2022).

Revision:

The revised sentences “The epitaxial growth mode with mixed crystalline orientations maintains a large strain transfer distance more than 90 nm in the superlattice films with lattice mismatch less than 1%. Such epitaxial strain permits sustainable long-range modulation of oxygen octahedral rotation and tilting, thereby inducing and regulating HIF.” were added in the Abstract (page 2/lines 8-12).

The revised sentence “Secondly, strain-induced structural distortions are quite complex and difficult in tailoring²⁴; and the epitaxial strain in thin films not only depends on the large lattice mismatch^{25, 26}, but also decays rapidly with increasing film thickness^{27, 28}.” was added in page 4/lines 14-17.

The new sentence “Notably, for determining HIF, the thickness-dependent epitaxial strain can maintain a single mode of octahedral distortion during strain regulation, which facilitates the demonstration of non-monotonic ferroelectric contributions from the coupling of OOR and OOT.” was added in page 4/lines 24-27.

The new sentence “More importantly, such thickness-dependent epitaxial strain in the films does not change the lattice mismatch, thus the stability of the oxygen octahedral distortion pattern in the sustainable long-range modulation can be greatly ensured.” was added in page 7/lines 24-27.

The new sentences “Generally, HIF can be theoretically predicted in A/B-site ordered double-perovskite superlattice systems, but the experimental difficulties in controlling epitaxial strain and octahedral distortion hinder the inducing of HIF in perovskite superlattices. For the thickness-dependent strain without changing the lattice

mismatch, the stability of octahedral distortion pattern in the sustainable long-range modulation is greatly ensured in LNMO/LCMO superlattices.” were added in page 15/lines 25-30.

The new sentence “In this work, we overcame the universal difficulties of the sustainable long-range modulation for octahedral distortion in designing HIF by regulating thickness-dependent epitaxial strain, and achieved a tunable HIF at room temperature in $\text{La}_2\text{NiMnO}_6/\text{La}_2\text{CoMnO}_6$ double-perovskite superlattices.” was added in page 4/lines 28-31.

The new sentences “A large strain critical thickness are achieved by the epitaxial growth mode with mixed crystalline orientations, which permits the sustainable long-range modulation of oxygen octahedral rotation and tilting. Such strain-driven octahedral distortion strategy in determining HIF, without changing the lattice mismatch, can effectively preclude serious interference from the changes of octahedral distortion pattern.” were added in page 5/lines 4-9.

The new sentences “For designing HIF in experiments, driving oxygen octahedral rotation/tilting usually requires large octahedral rotation phase mismatch between the films and substrates. Thus, epitaxial strain plays a very important role in maintaining and enhancing octahedral rotation. The thickness-dependent epitaxial strain maintains a single mode of octahedral distortion during strain regulation, which facilitates the demonstration of non-monotonic ferroelectric contributions, as a crucial feature for identifying HIF in experimental measurements, from the strain-modulated coupling of octahedral rotation and tilting.” were added in page 19/lines 8-16.

***Comment 1:** The evolution of the BO_6 octahedron tilting/rotation could reflect through the SAED (which is absence in the manuscript or supporting information) or the FFT images (which shows no evidence of the BO_6 octahedron in the Fig. 1b). It is possibly that the concepts of SAED and FFT are confused in the article.*

Response: Thanks for the reviewer’s professional comments. We have performed the SAED tests. According to the combination of the extinction law and the ordered close-packed structures, we supplemented the corresponding discussion of epitaxial structures from the diffraction pattern.

As the reviewer pointed out, the concepts of SAED and FFT are different. The SAED pattern is the reciprocal space image for lattice structures which is measured directly by transmission electron microscope (TEM) through rotating the selected diaphragm. Thus, the SAED pattern contains the separable information about the amplitude (intensity) of the electron wave. While the FFT image is derived from a mathematical transformation for high-resolution TEM image of the lattice, which contains the information both of amplitude and phase of the electron wave. Therefore, the FFT reflecting phase information from HR-TEM image can be used to study the computational simulations on structural image, such as GPA, whereas the SAED pattern cannot. In general, from the crystal structure point of view, the distributions of lattice planes reflected from SAED and FFT in the reciprocal space are the same, corresponding to the same set of diffraction spot patterns.

In this work, we use diffraction spot patterns to determine the epitaxial growth mode of the double-perovskite superlattices, which is consistent for FFT and SAED. As shown in Supplementary Figure S2, the SAED patterns indicate three growth modes of LNMO/LCMO superlattices grown on (001)-Nb:STO substrates, i.e., type-I, -II and -III; $001_{\text{SL}}/001_{\text{Sub}}$, $[001](110)_{\text{SL}}/[010](001)_{\text{Sub}}$ and $[001](110)_{\text{SL}}/[100](001)_{\text{Sub}}$, respectively. Notably, the additional diffraction spots of type-II and -III are more obvious than that of type-I, which indicates a relatively few proportions of type-I epitaxial structures within the tested region. These extra diffraction spots originate from the superstructures of *B*-site ordered double perovskites. In fact, the extinction law of space group $P2_1/n$ is $(h\ 0\ l)$ and $(0\ k\ 0)$, where $h + l = \text{odd number}$; $k = \text{odd number}$, respectively. For the type-I and -II, the simplest extra diffraction spot originates from $(00l)$, where l is odd number, because (001) is the close-packed plane for ordered *B*-site cations except the extinction of primary diffraction. While for the type-III, the emitted electron beam is along $[001]$ zone axis for generating diffraction, the second close-packed plane is $(0\ k\ 0)$. Thus, the simplest extra diffraction spot originates from $(0\ 1\ 0)$. These discussions have been added to the section of structure analysis both in main text and Supplementary Information. Furthermore, the good uniformity and stoichiometry of *B*-site cations in double perovskites are quite crucial for generating extra diffraction which is dependent on the close-packed plane with ordered *B*-site cations. We then tested EDS

images for superlattice films to indicate the good uniformity and stoichiometry for *B*-site cations. As shown in Supplementary Figure S3, the elemental distribution is homogeneous and the ratio of *B*-site atoms close to 2:1:1. The related analysis on EDS mapping was added in Supplementary Information.

In addition, since the rotation/tilting of the oxygen octahedra belongs to the fine evolution inside the crystal and the atom mass of oxygen is much too light, it is quite difficult to reflect octahedra rotations by lattice-scale diffraction spots measurements. By investigating the literature, we adopted annular bright-field (ABF) STEM images with atomic resolution to demonstrate the octahedral rotations in superlattice films, and the related results and discussions are shown in *Comment 3*.

Thanks for the reviewer's comments again. According to the reviewer's comments, we have added the necessary measurements and supplements on electron diffraction and epitaxial structures. The corresponding discussions and revisions greatly improve the quality of our manuscript.

Supplementary Figure S2. Selected area electron diffraction of LNMO/LCMO superlattices with different epitaxial modes. **a**, SAED pattern of double-perovskite superlattice films SL₉₀. **b**, The local magnification image of additional diffraction spots corresponding to three types of growth modes, marked by marked by circles and arrows in different colors, respectively. The additional diffraction spots of II- and III-type are more obvious than that of I-type, which indicates a relatively few proportions of I-type epitaxial structures within the tested region.

Supplementary Figure S3. The percentage of B-site element content for SL₉₀ superlattice films. **a**, A low-magnification BF-STEM image of SL₉₀ superlattice films on STO (001) substrate viewed along the [100] zone axis of the substrates. **b-g**, EDS-elemental mapping of Sr, Ti, O, Mn, Ni and Co for SL₉₀ films, respectively. **h**, Atom percentage of Mn, Co and Ni. **i**, EDS spectrum of SL₉₀. The elemental distribution is homogeneous, with the ratio of B-site atoms close to 2:1:1, indicating double-perovskite superlattices with good uniformity and stoichiometry for B-site cations.

Revision:

The new figures of SAED and EDS-mappings were added in Supplementary Information as Supplementary Figure S2 and S3, respectively.

The revised sentence “According to the extra spots highlighted by the white arrows in FFT patterns, three types of double-perovskite growth modes on Nb:STO substrates (type-I, -II and -III) are determined as $001_{SL} // 001_{Sub}$, $[001](110)_{SL} // [010](001)_{Sub}$ and $[001](110)_{SL} // [100](001)_{Sub}$, respectively.” was added in page 6/lines 17-20.

The new sentence “The same extra diffraction spots in SAED are systematically analyzed by a combination of the extinction law and the ordered close-packed structures (Supplementary Figs. S2 and S3).” was added in page 6/lines 23-25.

The new sentences “Notably, the additional diffraction spots of type-II and -III are more obvious than that of type-I, which indicates a relatively few proportions of type-I epitaxial structures within the tested region. These extra diffraction spots originate from the superstructures of B-site ordered double perovskites. In fact, the extinction law of space group $P2_1/n$ is $(h\ 0\ l)$ and $(0\ k\ 0)$, where $h + l = \text{odd number}$; $k \equiv \text{odd number}$, respectively. For the type-I and -II, the simplest extra diffraction spot originates from $(00l)$, where l is odd number, because (001) is the close-packed plane for ordered B-site cations except the extinction of primary diffraction. While for the type-III, the emitted electron beam is along $[001]$ zone axis for generating diffraction, the second close-packed plane is $(0\ k\ 0)$. Thus, the extra diffraction spot originates from $(0\ k\ 0)$, where k is odd number. Furthermore, EDS mappings were tested to indicate the good uniformity and stoichiometry of B-site cations in double perovskites. As shown in Supplementary Figure S3, the elemental distribution is homogeneous and the ratio of B-site atoms close to 2:1:1. These results contribute to generate extra diffraction spots from the close-packed plane with B-site orderings in superlattices.” were added in Supplementary Information page 1/line 22 to page 2/line 4.

The new sentence “High-resolution lattice images, SAED and EDS mappings were measured by a field emission transmission electron microscopy (TEM, JEOL, JEM-2100F).” was added in Methods (page 20/lines 26-28).

Comment 2: For the ε_{xx} used the vector \mathbf{g}_1 which is perpendicular to the interface, is it possible the Fig. 2b indicates the situation of out-of-plane strain? Or there may be some mistakes in the annotation of the images. Meanwhile, the stripe-like contrast in the grayscale of Fig. 2c should also be perpendicular to its reciprocal lattice vectors. Thus, all the GPA analysis need to be checked carefully.

Response: Thanks for the reviewer’s professional comments. We have carefully checked all the GPA analyses in our manuscript and apologize for the confusions caused by the inaccurate expressions and interpretations on GPA. The epitaxial strain within LNMO/LCMO superlattices are systematically analyzed by using the newest generation of GMS-3.5.3 with the FRWRtools plugin. According to the comments, we have responded to the questions and revised the manuscript. The details of the related

analysis and description of GPA are as follows.

As suggested by the reviewer, the correlation between strain field, phase contrast, and reciprocal lattice vectors should be unified and standardized with each other. For the accuracy of the strain analysis, we comprehensively investigated the basic principles and operating procedures of GPA to ensure this correlation. GPA is an effective approach for processing HRTEM images by combining real-space and Fourier-space information to estimate and visualize the spatial distribution of strain¹. By measuring the displacement of lattice fringes of HRTEM image with respect to an unstrained reference area, the local Fourier components of lattice fringes is calculated so that the information concerning the strain of lattice can be extracted by analyzing interference fringes¹. The GPA method is based upon centering an aperture around the assigned reflection in the Fourier transformation of an HRTEM image and subsequently performing an inverse Fourier transformation. The phase of image, namely geometric phase $\mathbf{Pg}(\mathbf{r})$, is related to the component of displacement field $\mathbf{u}(\mathbf{r})$ in the direction of the reciprocal lattice vector \mathbf{g} :

$$\mathbf{Pg}(\mathbf{r}) = -2\pi\mathbf{g} \cdot \mathbf{u}(\mathbf{r}) \quad (1)$$

where \mathbf{r} is the position in the image¹. A two-dimensional lattice is defined in real space basis vectors \mathbf{a}_1 and \mathbf{a}_2 , which correspond to the reciprocal lattice vectors \mathbf{g}_1 and \mathbf{g}_2 , respectively. Through calculating two sets of lattice fringes, the displacement field is given by¹:

$$\mathbf{u}(\mathbf{r}) = -(2\pi)^{-1}[\mathbf{Pg}_1(\mathbf{r}) \cdot \mathbf{a}_1 + \mathbf{Pg}_2(\mathbf{r}) \cdot \mathbf{a}_2] \quad (2)$$

The information of local strain can be obtained by analyzing the gradient of the displacement field, which is defined as $\boldsymbol{\varepsilon}$ and described as matrix form:

$$\begin{aligned} \boldsymbol{\varepsilon} &= \begin{pmatrix} \varepsilon_{xx} & \varepsilon_{xy} \\ \varepsilon_{yx} & \varepsilon_{yy} \end{pmatrix} \\ &= \begin{pmatrix} \partial_{ux}/\partial_x & \partial_{ux}/\partial_y \\ \partial_{uy}/\partial_x & \partial_{uy}/\partial_y \end{pmatrix} \end{aligned} \quad (3)$$

In the above matrix, the value of each element can be converted to an image. More details about GPA method can be found in Hýtch's work¹. All in all, in an ideal situation, the selected reciprocal lattice vector is perpendicular to the orientation of crystal planes in real space, and the contrast of geometric phase is perpendicular to the reciprocal lattice vector. In fact, however, lattice defects, deviations of zone axes, multiphase coexistence, as well as different reference areas, can cause significant

changes in the strain field and geometrical phases. Thus, during the operation for GPA, two non-colinear reciprocal space vectors (\mathbf{g}_1 and \mathbf{g}_2) with large intensities of power spectra should be selected to reach an excellent signal-to-noise ratio². Furthermore, a proper mask size in Fourier space is regulated to smooth the image around the peaks and reduce the initial noise during Fourier filtering².

As shown in Supplementary Figure S8, the calculated GPA displays a standard correlation between strain field (ε_{xx} and ε_{yy}), phase contrast (\mathbf{Pg}_1 and \mathbf{Pg}_2) and reciprocal lattice vectors (\mathbf{g}_1 and \mathbf{g}_2). Epitaxial strain is released through the dislocations and lattice distortions along the boundaries of different regions. Notably, the phase contrast is significantly intermittent, indicating the influence caused by the lattice defects in the superlattice films. Similarly, Figure 2a-2c behaves a standard geometric correlation in GPA calculation for the TEM image of the SL₃₀ at interface. The strain fields are well visualized at the macroscopic lattice scale. Response Figure 1 presents the full primary files and operating panel for the GPA calculation. However, as shown in Figure 2d-2f, the GPA calculation of the film surface is slightly different. The phase contrasts (\mathbf{Pg}_1 and \mathbf{Pg}_2) are not perfectly perpendicular to the reciprocal lattice vectors (\mathbf{g}_1 and \mathbf{g}_2). This is attributed to the fact that, at the atomic scale, the atomic columns corresponding to the multiple epitaxial modes do not form a single focused diffraction spot, but are accompanied by satellite peaks, which are reflected in the macroscopic expression of the phase contrasts. In addition, to some extent, it may also be related to the oxygen octahedral distortion. Even so, it should be reminded that this phase changes do not affect the results of the strain analysis, which is mainly carried out in only one phase period (from $-\pi$ to π) and the atoms in the reference region are well aligned. The standard operation and corresponding primary files for GPA are presented in Response Figure 2. We estimated the local strain using a statistical approach that corresponds to the epitaxial structure (Supplementary Figure S7).

We have updated our GPA calculations, and the epitaxial strain were thoroughly clarified and discussed in the manuscript. Thanks for the reviewer's professional comments again. These revisions make the strain visualization more accurate and greatly improve the quality of our manuscript.

Supplementary Figure S8. Geometric phase analysis and lattice dislocations of the SL_{90} films. **a,b**, HRTEM image of the SL_{90} (**a**) and the corresponding FFT image (**b**). **c**, The FFT images of the selected areas in (**a**). The interplanar distances of OP change more obviously than that of IP, which indicates the release of local epitaxial strain with film thickness. **d**, The monochromatic filtered IFFT images of IP and OP. The IP lattice fringes are regular, while the OP lattice fringes show considerable areas with T-type dislocations and lattice deformations. **e-g**, GPA analysis of phase images (**e**) and IP strain ϵ_{xx} (**f**) and OP strain ϵ_{yy} (**g**). The non-collinear reciprocal lattice vectors \mathbf{g}_1 and \mathbf{g}_2 are selected by white circles in FFT image (**b**) for GPA. Yellow square in (**a**) is the reference region for GPA. The local strains are concentrated at the boundaries of regions (1), (2) and (3), which is consistent with the discontinuity of the output phase. These strains are originated from the dislocations in the epitaxial direction since the clamping action of the substrates decreases as the thickness.

Fig. 2 | Geometric phase analysis (GPA) and octahedral distortions of the LNMO/LCMO superlattice films. **a** Low magnification TEM image of the SL₃₀ at interface and the FFT image of the films. **b, c** Corresponding GPA analysis of (a) along in-plane direction ϵ_{xx} (b) and out-of-plane direction ϵ_{yy} (c), respectively. **d** HAADF STEM image of the SL₆₀ near surface and the inset of FFT image. **e, f** Corresponding GPA analysis of ϵ_{xx} (e) and ϵ_{yy} (f) on local atomic image (d), respectively. The insets attached to ϵ_{xx} and ϵ_{yy} are corresponding phase images with normalized phase variation from $-\pi$ to π (black to white). White circles in FFT images mark the non-collinear reciprocal lattice vectors \mathbf{g}_1 and \mathbf{g}_2 for GPA. Yellow squares show the reference region for GPA. The color scale indicates the relative difference of local strain in the films. **g** Average intensity profiles of the red and blue lines in the grayscale images of ϵ_{xx} (e) and ϵ_{yy} (f), respectively. **h** Local annular bright-field (ABF) STEM images of the SL₆₀ films and STO. The schematic shows the corresponding BO_6 octahedral distortions. **i** The ABF-STEM image of the cross-sectional interface of SL₆₀ on STO and the tilting angle ($B-O-B'$) of oxygen octahedrons by collecting 19 layers of perovskite unit cells.

Supplementary Figure S7. Geometric phase analysis (GPA) of the LNMO/LCMO superlattice films. a, STEM image of local regions at higher magnification for Figure 2d. **b,** The IP and OP atomic displacement fields (u_{xx} and u_{yy} , respectively) of local regions framed in green in (a). The green and red spheres represent *A*-site and *B*-site cations, respectively. The local atomic images with neat.

Response Figure 1. The process of geometric phase analysis on the interface of SL_{30} films performed by using the FRWRtools plugin in GMS-3.5.3. Two non-collinear reciprocal lattice vectors \mathbf{g}_1 and \mathbf{g}_2 are selected for GPA. The IP and OP strains are output as ϵ_{xx} and ϵ_{yy} , respectively. The corresponding strain and phase variations are visualized by the calibration scales from -1% to 1% and from $-\pi$ to π , respectively. The resolution and smoothing are set to 1 nm and 10, respectively. The final analysis files include diffraction image, and IP and OP displacement fields (u_{xx} and u_{yy}), strain fields (ϵ_{xx} , ϵ_{yy} , ϵ_{xy} and rotation) and phase (P_{g1} and P_{g2}).

Response Figure 2. The process of GPA on the surface of SL_{30} films performed by using the FRWRtools plugin in GMS-3.5.3. Two non-collinear reciprocal lattice vectors \mathbf{g}_1 and \mathbf{g}_2 are selected for GPA. The IP and OP strains are output as ϵ_{xx} and ϵ_{yy} , respectively. The corresponding strain and phase variations are visualized by the calibration scales from -1% to 1% and from $-\pi$ to π , respectively. The resolution and smoothing are set to 1 nm and 10, respectively. The final analysis files include diffraction image, and IP and OP displacement fields (u_{xx} and u_{yy}), strain fields (ϵ_{xx} , ϵ_{yy} , ϵ_{xy} and rotation) and phase (P_{g1} and P_{g2}).

Response References:

1. M. J. Hÿtch, E. Snoeck & R. Kilaas. Quantitative measurement of displacement and strain fields from HREM micrographs. *Ultramicroscopy* **74**, 131-146 (1998).
2. J. Huang, et al. Mechanism of Sc poisoning of Al-5Ti-1B grain refiner. *Scripta Mater.* **180**, 88-92 (2020).

Revision:

The new figures of GPA (Figure 2a-2g and Supplementary Figure S7 and S8) were updated in Main Text and Supplementary Information, respectively.

The new sentences “However, the phase contrasts (P_{g1} and P_{g2}) are not perfectly perpendicular to the reciprocal lattice vectors (\mathbf{g}_1 and \mathbf{g}_2). This is attributed to the diffraction spots with satellite peaks raised from the atomic columns under the

multiple epitaxies, reflecting in the macroscopic expression of the phase contrasts. Even so, it should be noted that this phase reflection does not affect the results of the strain analysis because it is mainly carried out in only one phase period (from $-\pi$ to π) and the atoms in the reference region are well aligned.” were added in page 9/lines 11-17.

The new sentences “The calculated GPA displays a standard geometric correlation between strain field (ε_{xx} and ε_{yy}), phase contrast (Pg_1 and Pg_2) and reciprocal lattice vectors (g_1 and g_2). The FFT of three different regions clearly demonstrates the changing process of the lattice parameters. Epitaxial strain is released through the dislocations and lattice distortions along the boundaries of different regions. Notably, the phase contrast is significantly intermittent, indicating the influence caused by the lattice defects in the superlattice films.” were added in Supplementary Information page 4/lines 6-12.

The new sentences “We utilized an ozone-assisted growth method to achieve the growth of high-quality double-perovskite superlattices. As shown in Supplementary Figure S6, the local HAADF images reveal fully epitaxial structures with the coherent growth for the LNMO and LCMO superlattice layers. Moreover, for the local IP and OP displacement fields (u_{xx} and u_{yy}) of atomic columns, the displacement distributions of *A*-site cations are almost unchanged along the IP and OP directions, while changing slightly for *B*-site cations. This difference in the atomic projection, to some extent, indicates the tilting/rotation of BO_6 octahedron in the superlattice films, which can usually affect the ferroic order parameters in the system. The reliable rotation or tilting of the oxygen octahedron is analyzed by the changes in *B-O-B* bond angles, according to the ABF STEM images for oxygen distribution (see Figure 2h). Supplementary Figure S7 shows the strain maps processed in grayscale to reflect the strain distribution clearly. According to the practical in-plane and out-of-plane crystal axes, the appropriate cutting lines were employed to obtain the intensity profiles of strain.” were added in Supplementary Information page 3/lines 1-15.

Comment 3: *The conclusion of the tilting/rotation of BO_6 octahedron in the superlattice films made by Fig. 2h need to be reconsidered, at least providing the quantitative data. The discrepancy of the displacement is difficult to observe only by the individual atomic-scale HAADF-STEM image.*

Response: Thanks for the reviewer's constructive comments on the quantitative analysis of oxygen octahedral rotation/tilting. We agree with the reviewer that the atomic displacement of BO_6 octahedral distortions is difficult to be observed only by the individual atomic-scale HAADF-STEM image and the necessary quantitative analysis should be provided. The quantitative analysis of octahedral rotation/tilting can be performed from the detected distributions of oxygen coordination by using the STEM annular bright-field (ABF) technique which is sensitive to the atoms with light masses such as oxygen⁴².

As shown in Fig. 2h, the oxygen atomic columns of STO in cross-sectional ABF-STEM image are arranged in straight chains, which is consistent with the 180° bond angle of Ti-O-Ti in $Pm-3m$ symmetry. While for the LNMO/LCMO superlattices, the elongated oxygen sublattices are arranged in a zigzag-like pattern. This sharp contrast in the oxygen coordination indicates the obvious OOR/OOT in SL_{60} films. Compared to STO substrates with a single crystalline orientation, the partial oxygen sublattices in superlattices are relatively ambiguous due to the phase interfering from multiple growth modes. Since the apical O atom overlaps with A -site La atom, we can only determine the bond angles ($B-O-B'$) associated with OOR by measuring the oxygen positions on the left and right sides of B -site atoms. The quantitative bond angles of $B-O-B'$ are counted layer by layer within a large region as shown in Fig. 2i. The fluctuation of the bond angle fully reflects the modulation of the octahedral rotation by the epitaxial strain. In addition, the dramatic changes within the six perovskite layers exhibit the clamping effect of the STO substrates on OOR. This limitation of octahedral structures originates from the lattice and symmetry mismatch between the films and substrates, remaining a coherent interface and lattice connectivity⁴³. According to the approximately equal values of median and average of bond angles within a single layer, OOR tends to be stable as the clamping effect disappears. The corresponding bond angle is estimated as $\sim 157^\circ$ on average, i.e., the octahedron is rotated by approximately 11.5° . The distribution sites of oxygen atom

columns are displayed in Response Figure 1 for counting the $B-O-B'$ bond angles, and the statistical data of $B-O-B'$ bond angles are listed in Response Table 1. In brief, the epitaxial strains in LNMO/LCMO superlattices have the ability to achieve the sustainable long-range modulation of the oxygen octahedral distortions, especially for OOR/OOT. The related discussions are added to the section of structure analysis.

Notably, the ABF image recorded along the STO-[110] axis is the best visualization to analyze OOR. Supplementary Figure S10 shows annular bright-field (ABF) STEM images of the SL_{60} films along different crystal axis directions of SL-[110] and $[-0 -1 1]$. According to the corresponding schematic of oxygen distribution, the layered structure of oxygen columns measured along SL-[110] axis cannot be observed clearly. The stacking of A - and B -site atoms in close-packed plane does not provide the sufficient atomic gaps for the oxygen atoms with lighter masses to reflect the contrast of the sublattice projections. Consequently, despite the fact that the octahedron is in a simple geometric perspective, the tilting or rotation of the oxygen octahedron cannot be determined. While, along SL-[0 -1 1] axis, the elongated and misaligned oxygen sublattices are arranged in a zigzag-like pattern. This result significantly indicates the qualitative octahedral distortions in the superlattice films. The layered structure of oxygen columns forms a close-packed plane which is highlighted both in the ABF image and structural schematic by the red dashed line and blue rectangular box, respectively. However, the quantitative analysis of BO_6 octahedral distortions cannot be proceeded because the geometrical position of octahedron is too complex to measure any bond angles of $B-O-B'$ for OOR/OOT. Supplementary Figure S11 shows the illustrations of the geometric structure for the sample preparation by using focused ion beam (FIB) and the STO-[110] axis for the ABF measurements. The films are sliced along the diagonal of the STO substrates ($\sim 45^\circ$) for FIB preparation. The distribution features of oxygen columns for both superlattice films and STO substrates are clearly displayed in the schematic along the STO-[110] axis for ABF measurements. More importantly, the quantitative analysis of the BO_6 distortions can be further performed by measuring the bond angle of $B-O-B'$

under a simple geometrical perspective of oxygen octahedron. The basic information of the superlattice films and FIB sample for ABF-STEM measurement are shown in Supplementary Figure S12. The related ABF-STEM analysis is added to the Section I of Supplementary Information.

Thanks for the reviewer's comments again. The corresponding additions and analyses on ABF-STEM measurements further indicate the strain-driven oxygen octahedral distortion in LNMO/LCMO superlattices. More importantly, the quantitative analysis of OOR firmly supports the conclusion of long-range modulated octahedrons and provides direct evidence for the inducing and identifying of hybrid improper ferroelectricity (HIF). The manuscript has been greatly improved.

Fig. 2 | Geometric phase analysis (GPA) and octahedral distortions of the LNMO/LCMO superlattice films. **h** Local annular bright-field (ABF) STEM images of the SL₆₀ films and STO. The schematic shows the corresponding BO₆ octahedral distortions. **i** The ABF-STEM image of the cross-sectional interface of SL₆₀ on STO and the tilting angle ($B-O-B'$) of oxygen octahedrons by collecting 19 layers of perovskite unit cells.

Supplementary Figure S10. Annular bright-field (ABF) STEM images of the SL_{60} films along different crystal axis directions. a,b, Local ABF-STEM image of SL_{60} along $[110]$ direction (a) and the corresponding schematic of oxygen distribution (b). Due to the stacking of A - and B -site atoms with large masses, the close-packed layer structure of oxygen column is not observed in $[110]$ direction for the SL films. Consequently, the tilting or rotation of the oxygen octahedron cannot be determined. **c,d**, ABF-STEM image with oxygen column contrast along $[0 -1 1]$ direction for SL_{60} (c) and the corresponding oxygen distribution (d). The misaligned oxygen atoms highlighted by the curved red dashed line qualitatively indicate the oxygen octahedral distortions in the films. However, the quantitative analysis of this distortion cannot be proceeded due to the geometrical complexity of the octahedral position.

Supplementary Figure S11. The illustrations of the geometric structure for the sample preparation by using focused ion beam (FIB) and the crystal axis for the ABF measurements. The film samples are sliced parallel to the diagonal of the STO substrates ($\sim 45^\circ$) during the FIB preparation. Along the axis of STO-[110], the tested ABF images can clearly show the distribution feature of oxygen columns for both SL films and STO substrates. The misaligned oxygen atoms can indicate the BO_6 octahedral distortions in the films and the quantitative analysis of the distortions can be further proceeded by measuring the bond angle of $B-O-B'$ under a simple geometrical position of oxygen octahedron.

Supplementary Figure S12. The basic information of the superlattice films and FIB sample for ABF-STEM measurement. **a**, The superlattice films grown on (001)-Nb:STO with the size of $5 \times 5 \text{ mm}^2$. **b**, Electron microscopy images of FIB samples with two thin areas at low magnification. **c**, The cross-section of SL_{60} on Nb:STO. The jagged shape of the film surface is damage zone caused by the ion beam during the FIB-sample preparation.

Response Figure 1. The distribution sites of oxygen atom column for the statistics of the $B-O-B'$ bond angles.

Response Table 1. The statistical bond angles of $B-O-B'$ in the ABF-STEM image.

Layer number	Bond angle (°)																	
1	180	180	180	180	180	180	180	180	180	180	180	180	180	180	180	180	180	
2	180	180	180	180	180	180	180	180	180	180	180	180	180	180	180	180	180	
3	180	180	180	180	180	180	180	180	180	180	180	180	180	180	180	180	180	
4	160.0	157.0	155.2	163.6	162.9	155.3	150.9	152.2	149.9	143.9	148.9	152.4	147.7	146.6	148.0	150.7	157.9	159.9
5	154.1	151.4	153.6	162.9	164.7	152.6	148.3	151.4	153.4	153.6	149.7	150.5	147.9	145.8	152.4	156.4	164.6	167.7
6	157.8	151.9	151.3	149.9	151.6	156.5	153.8	147.1	149.2	154.9	151.6	150.3	159.8	162.7	156.9	158.2	166.0	168.2
7	153.9	159.7	160.6	150.7	145.6	143.0	144	150.4	148.0	147.0	154.9	156.7	155.5	152.4	153.1	154.2	144.8	133.3
8	153.8	160.1	158.1	150.1	152.8	150.5	144.6	147.0	152.6	152.8	153.0	159.7	162.5	157.8	157.8	160.1	157.5	153.3
9	154.1	156.3	143.2	135.9	143.2	148.5	147.7	147.4	150.8	152.0	152.3	146.8	146.9	152.7	148.5	148.6	157.8	161.9
10	147.1	147.9	151.6	149.1	144.7	148.9	155.1	157.2	153.9	157.6	164.4	162.1	153.3	146.1	147.6	151.6	150.3	152.4
11	152.6	149.1	156.1	157.2	156.3	157.5	157.8	157.0	155.8	159.6	164.1	159.8	159.2	157.6	149.6	150.0	153.4	149.4
12	159.7	157.2	156.3	152.4	158.3	161.8	155.5	150.5	150.0	158.5	155.9	149.6	152.5	154.3	154.4	151.3	150.2	152.3
13	158.4	157.1	155.0	157.5	164.2	163.4	153.3	150.3	160.8	157.2	149.4	151.8	151.3	156.8	160.0	155.6	158.0	162.6
14	156.9	159.6	159.8	154.3	148.7	148.2	151.8	152.5	157.4	159.6	152.5	147.8	153.0	160.6	155.1	146.8	149.8	156.4
15	150.0	151.0	149.7	145.8	142.4	148.8	163.1	165.8	163.6	162.0	158.9	155.2	154.4	154.6	150.7	152.6	158.8	154.1
16	161.4	158.6	162.9	153.0	152.1	160.8	153.9	145.6	147.7	150.4	150.0	153.8	156.7	155.8	151.7	144.0	146.4	155.3
17	152.2	152.7	155.4	151.6	150.0	152.1	154.3	152.6	155.4	160.4	158.4	158.2	159.7	158.9	160.2	159.9	154.7	151.4
18	165.2	157.1	150.9	153.2	153.2	152.5	151.7	152.7	155.6	150.8	149.8	145.2	143.3	150.7	146.5	144.2	154.6	159.5
19	149.8	151.7	156.9	154.6	153.0	145.9	147.4	151.6	144.3	144.3	148.3	153.2	161.3	158.5	151.3	157.1	162.2	160.7

Reference:

42. Ishikawa, R. et al. Direct imaging of hydrogen-atom columns in a crystal by annular bright-field electron microscopy. *Nat. Mater.* **10**, 278–281 (2011).
43. X. Ding, et al. Crystal symmetry engineering in epitaxial perovskite superlattices. *Adv. Funct. Mater.* **31**, 2106466 (2021).

Revision:

The new figures of ABF-STEM images (Figure 2h and 2i, Supplementary Figure S6 and S10-S12) were added in Main Text and Supplementary Information, respectively.

The new sentence “High-angle annular dark-field (HAADF) and annular bright-field (ABF) STEM measurements were carried out using a CEOS Cs-corrector operated at 200kV (Thermo Scientific, Themis Z).” was added in Methods (page 20/lines 28-30).

The new sentences “Epitaxial strain in \$ABO_3\$ perovskites can typically induce \$BO_6\$ octahedral distortions or change the lattice symmetry^{40, 41}. It is more specific for effecting OOR/OOT in double perovskites with naturally low symmetry. The annular bright-field (ABF) STEM measurement, which is sensitive to light atoms such as oxygen⁴², was then performed to visualize \$BO_6\$ octahedral distortion in the strained \$SL_{60}\$ films. As shown in Fig. 2h, the oxygen atomic columns of STO in cross-sectional ABF-STEM image are arranged in straight chains, which is consistent with the \$180^\circ\$ bond angle of Ti-O-Ti in \$Pm-3m\$ symmetry. While for the LNMO/LCMO superlattices, the elongated oxygen sublattices are arranged in a zigzag-like pattern. This sharp contrast in the oxygen coordination indicates the obvious OOR/OOT in \$SL_{60}\$ films. Compared to STO substrates with a single crystalline orientation, the partial oxygen sublattices in superlattices are relatively ambiguous due to the phase interfering from multiple growth modes. Since the apical O atom overlaps with \$A\$ -site La atom, we can only determine the bond angles (\$B-O-B'\$ ) associated with OOR by measuring the oxygen positions on the left and right sides of \$B\$ -site atoms. Notably, the ABF image recorded along the STO-[110] axis is the best visualization to analyze OOR (Supplementary Figs. S10-S12). The quantitative bond angles of \$B-O-B'\$ are counted layer by layer within a large region as shown in Fig. 2i. The fluctuation of the bond angle fully reflects the modulation of the octahedral rotation by the epitaxial strain. In addition, the dramatic changes within the six perovskite layers exhibit the clamping”

effect of the STO substrates on OOR. This limitation of octahedral structures originates from the lattice and symmetry mismatch between the films and substrates, remaining a coherent interface and lattice connectivity⁴³. According to the approximately equal values of median and average of bond angles within a single layer, OOR tends to be stable as the clamping effect disappears. The corresponding bond angle is estimated as $\sim 157^\circ$ on average, i.e., the octahedron is rotated by approximately 11.5° . Therefore, the epitaxial strains in LNMO/LCMO superlattices have the ability to achieve the sustainable long-range modulation of the oxygen octahedral distortions, especially for OOR/OOT.” were added in page 9/line 26 to page 10/line 25.

The new sentence “Compared with proper ferroelectrics, such intrinsic weak ferroelectricity is improper in origin, because epitaxial strain usually causes OOR/OOT in double perovskites as confirmed in Fig. 2.” was added in page 13/lines 21-24.

The new sentence “The average of calculated bond angles for SL_{60} is similar to the value measured in ABF-STEM image, indicating the reliability of the calculations.” was added in page 17/line 30 to page 18/line 2.

The new sentences “Supplementary Figure S10 shows annular bright-field (ABF) STEM images of the SL_{60} films along different crystal axis directions of SL -[110] and SL -[0 -1 1]. According to the corresponding schematic of oxygen distribution, the layered structure of oxygen columns measured along SL -[110] axis cannot be observed clearly. The stacking of A - and B -site atoms in close-packed plane does not provide the sufficient atomic gaps for the oxygen atoms with lighter masses to reflect the contrast of the sublattice projections. Consequently, despite the fact that the octahedron is in a simple geometric perspective, the tilting or rotation of the oxygen octahedron cannot be determined. While, along SL -[0 -1 1] axis, the elongated and misaligned oxygen sublattices are arranged in a zigzag-like pattern. This result significantly indicates the qualitative octahedral distortions in the superlattice films. The layered structure of oxygen columns forms a close-packed plane which is

highlighted both in the ABF image and structural schematic by the red dashed line and blue rectangular box, respectively. However, the quantitative analysis of BO_6 octahedral distortions cannot be proceeded because the geometrical position of octahedron is too complex to measure any bond angles of $B-O-B'$ for OOR/OOT.” were added in Supplementary Information page 4/line 19 to page 5/line 4.

The new sentences “The best visualization of recording ABF-STEM image to analyze OOR is along the STO-[110] axis, i.e., SL-[100] or [010] axes. Supplementary Figure S11 shows the illustrations of the geometric structure for the sample preparation by using focused ion beam (FIB) and the STO-[110] axis for the ABF measurements. The films are sliced along the diagonal of the STO substrates ($\sim 45^\circ$) for FIB preparation. The distribution features of oxygen columns for both superlattice films and STO substrates are clearly displayed in the schematic along the STO-[110] axis for ABF measurements. More importantly, the quantitative analysis of the BO_6 distortions can be further performed by measuring the bond angle of $B-O-B'$ under a simple geometrical perspective of oxygen octahedron. The basic information of the superlattice films and FIB sample for ABF-STEM measurement are shown in Supplementary Figure S12.” were added in Supplementary Information page 5/lines 6-16.

Added Reference:

40. S. Li, et al. Strong ferromagnetism achieved via breathing lattices in atomically thin cobaltites. *Adv. Mater.* **33**, 2001324 (2020).
42. Ishikawa, R. et al. Direct imaging of hydrogen-atom columns in a crystal by annular bright-field electron microscopy. *Nat. Mater.* **10**, 278–281 (2011).
43. X. Ding, et al. Crystal symmetry engineering in epitaxial perovskite superlattices. *Adv. Funct. Mater.* **31**, 2106466 (2021).

Finally, thanks to the editor and reviewers again for all the excellent and professional comments, which give us a great chance to improve our work better.

REVIEWER COMMENTS

Reviewer #1 (Remarks to the Author):

This work presents some interesting and important results, and the authors have revised it well.

Reviewer #2 (Remarks to the Author):

There are still some concerns which need to be addressed before the publication in this high-quality journal:

- 1) The authors should check the use of “in-plane” and “out-of-plane” again to avoid the misleading of the readers.
- 2) Does there exist OOR or OOT in the bulk of $\text{La}_2\text{NiMnO}_6$ or $\text{La}_2\text{CoMnO}_6$?
- 3) The tool/method of simulating the oxygen positions should be added in the part of methods in the article.

Below is a list of the main changes to the manuscript:

- (1) Results: We defined the directions of “in-plane” (IP) and “out-of-plane” (OP).
- (2) Discussion: The analysis of HIF in LNMO/LCMO double-perovskite superlattices was further emphasized.
- (3) References: Three highly relevant references were added to expound the mechanism of HIF.
- (4) Methods: The details of determining oxygen positions for BO_6 octahedral distortions are added.

In the following, the reviewer’s original comments are shown by black italic characters. Our Responses are shown by blue characters, and Revisions are shown by red characters.

Response to Reviewers' Comments

We appreciate the reviewers' time and effort for reviewing our manuscript again. The reviewers' valuable comments and suggestions were very professional and helped us to further improve our manuscript. The following is point-to-point response to their comments. We have carefully addressed all the questions and made necessary supplements in the revised manuscript. All the corresponding revisions in the manuscript are highlighted.

Reviewer #1

Comments: *This work presents some interesting and important results, and the authors have revised it well.*

Response: Thanks for the reviewer's positive comments. We appreciate the reviewers' time and effort again for improving our manuscript.

Reviewer #2

Comments: *There are still some concerns which need to be addressed before the publication in this high-quality journal.*

Response: Thanks for the reviewer's comments. According to these professional comments, we have responded to all the questions and revised our manuscript carefully. All the use of "in-plane" and "out-of-plane" were checked in the manuscript. We highlighted the existing OOT and OOR and explained the reason they cannot induce HIF in the bulk of LNMO and LCMO, furthermore, the analysis of HIF in LNMO/LCMO superlattices has been emphasized in Discussion. The determination of oxygen positions have been also added in Methods. The point-to-point responses are as follows.

Comment 1: *The authors should check the use of "in-plane" and "out-of-plane" again to avoid the misleading of the readers.*

Response: Thanks for the reviewer's valuable comments. We apologize for the present misleading of "in-plane" and "out-of-plane" for readers due to the lack of corresponding definitions in our manuscript. In this work, we define the direction parallel to the film surface as the in-plane direction (IP) and the direction perpendicular to the film surface as the out-of-plane direction (OP). These definitions are convenient for us to study the role of macroscopic strain on regulating HIF in LNMO/LCMO superlattices with multimodal epitaxial growth. We have checked all the use of "in-plane" and "out-of-plane" in the manuscript.

According to the lattice mismatch between films and substrates in Supplementary Fig. S4, the epitaxial superlattice films are subjected to in-plane tensile strain because the IP lattice parameters of STO are larger than that of unstrained LNMO and LCMO. Thus, in the ideal case of constant cell volume, the superlattice films are subjected to OP compressive strain. The macroscopic and microscopic properties of IP tensile strain and OP compressive strain are analyzed by X-ray diffraction characterization and TEM measurement, respectively. 2θ -scans XRD shows that the OP interlayer spacing is compressive and gradually increases with the film thickness (Fig. 1c). Such macroscopic OP compressive strain and IP tensile strain are further

available from Q_z and Q_x in RSM tests, respectively (Fig. 1d). Microscopically, the distribution of local strains is visualized by GPA, according to the HR-TEM images (Fig. 2 and Supplementary Fig. S8). The changes of OP and IP interlayer spacing indicate the microscopic strain relaxations within the superlattice films (Supplementary Fig. S9). We have added well-defined terms for “in-plane (IP)” and “out-of-plane (OP)” in the manuscript.

Thanks again for the reviewer’s suggestion. The related revisions have effectively avoided the misleading of the readers on “in-plane” and “out-of-plane”.

Revision:

The revised sentences “Figure 1c exhibits the XRD θ - 2θ scans of superlattice films with different thicknesses. The macroscopic out-of-plane (OP) epitaxial strain of thin films is usually determined by this diffraction collected perpendicular to the film surface.” were added in page 7/lines 5-7.

The revised sentences “The reciprocal space mapping (RSM) around the asymmetric reflection $(103)_{pc}$ further determines the macroscopic epitaxial strain and the dependence of strain transfer on film thicknesses. As shown in Fig. 1d, Q_x of the SL_{10} for the direction parallel to the film surface, i.e. in-plane (IP), is aligned with that of the substrates, which indicates the coherent epitaxy of superlattices with full strain. According to the lattice parameters calculated from the Q_z (OP) and Q_x (IP), the superlattice films behave macroscopic IP tensile and OP compressive strain, and the strain relaxation occurs obviously due to the increasing film thickness.” were added in page 8/lines 1-8.

Comment 2: Does there exist OOR or OOT in the bulk of La_2NiMnO_6 or La_2CoMnO_6 ?

Response: Thanks for the reviewer’s professional comments. As mentioned in the manuscript (page 18/line 18), the BO_6 octahedron in the bulk of LNMO and LCMO does rotate or tilt indeed, which can be indexed as $a^-a^-c^+$ pattern in Glazer’s notation⁹.

However, this rotation pattern is possible to generate HIF only in the perovskite superlattice systems with cation orderings rather than in single bulk materials, i.e., the “chemical criterion” for breaking inversion symmetry⁵³.

As reported, the rotation pattern of $a^-a^-c^+$ satisfies “energy criterion” for achieving HIF^{9, 10, 53}. For further breaking inversion symmetry, we designed epitaxial strain-modulated octahedral distortions, including OOR/OOT and JT distortion, thereby realizing room-temperature HIF in LNMO/LCMO double-perovskite superlattices. The DFT calculations systematically explore the effect of epitaxial strain on the octahedral distortion mode (Fig. 4c to 4e). Furthermore, ABF-STEM images of SL₆₀ indicate that the octahedron is rotated by 11.5° (Fig. 2h and 2i), which is approximately consistent with the average results of B -O- B bond angles calculated by DFT for SL₆₀. These results sufficiently demonstrate that the octahedral distortions are effectively modulated by the epitaxial strain, which had been emphasized throughout the manuscript.

In addition, HIF is widely accepted to normally occur in the $ABO_3/A'BO_3$ perovskite superlattices rather than $ABO_3/AB'O_3$ systems due to the chemical criterion and energy criterion^{9, 11, 53}. For LNMO/LCMO superlattices with B -site cation orderings, it seems to be contrary to the chemical criterion of HIF. However, this double-perovskite superlattice with $a^-a^-c^+$ rotations is much more different from the general systems of $ABO_3/AB'O_3$. Under epitaxial strain, various oxygen octahedrons (MnO_6 , CoO_6 and NiO_6) will develop different degrees of rotation/tilting and JT distortion due to the different chemical properties, resulting in the different inversion symmetry of B -site positions. Furthermore, the multimodal epitaxial growth provides a relatively complex lattice environment, driving a polar trilinear term similar to that in the A/A' ordered system. Recently, the B -site ordered perovskite superlattice systems have been increasingly focused on the theoretical study of hybrid ferroelectricity and multiferroics. The combination of the ferroelectricity induced by octahedral rotation and the ferromagnetism/ferrimagnetism caused by the ordered arrangement of different magnetic ions is also calculated in the B -site ordered double-perovskite bilayer¹⁰. The coexistence of lattice distortions and charge transfer can induce the hybrid ferroelectricity or polar structures in the perovskite superlattices $(ABO_3)_n/(AB'O_3)_n$ ($n=2$)^{54, 55}. Overall, based on our results in LNMO/LCMO

superlattices, it seems that the double-perovskite superlattice systems of $(A_2BB'O_6)_n/(A_2BB''O_6)_n$ ($n=1$) could be a novel platform for achieving HIF through the unique structural design. The related analysis of HIF in *B*-site ordered perovskite superlattices has been further highlighted in Discussion.

Thanks for the reviewer's constructive comments again. We explained the reason why $a^-a^+c^+$ octahedral rotations cannot induce HIF in the bulk of LNMO and LCMO, and the analysis of HIF in LNMO/LCMO superlattices has been further emphasized in Discussion. These revisions greatly improve the quality of the article on HIF research.

References:

9. H. J. Zhao, et al. Near room-temperature multiferroic materials with tunable ferromagnetic and electrical properties. *Nat. Commun.* **5**, 4021 (2014).
10. J. Zhang, et al. Design of Two-dimensional multiferroics with direct polarization-magnetization coupling. *Phys. Rev. Lett.* **125**, 017601 (2020).
11. X. Shen, F. Wang, X. Lu & J. Zhang. Two-dimensional multiferroics with intrinsic magnetoelectric coupling in *A*-site ordered perovskite monolayers. *Nano Lett.* **23**, 735-741 (2022).
53. J. M. Rondinelli & C. J. Fennie. Octahedral rotation-induced ferroelectricity in cation ordered perovskites. *Adv. Mater.* **24**, 1961-1968 (2012).
54. H. Zhang, Y. Weng, X. Yao & S. Dong. Charge transfer and hybrid ferroelectricity in $(YFeO_3)_n/(YTiO_3)_n$ magnetic superlattices. *Phys. Rev. B* **91**, 195148 (2015).
55. Y. Weng, J. Zhang, B. Gao & S. Dong. $(LaTiO_3)_n/(LaVO_3)_n$ as a model system for unconventional charge transfer and polar metallicity. *Phys. Rev. B* **95**, 155117 (2017).

Revision:

The revised sentence “For the bulk of LNMO and LCMO, the ground-state structures of $P2_1/n$ symmetry possess antipolar displacements resulting from the octahedral rotation pattern of $(a^-a^+c^+)^9$.” was added in page 18/lines 18-20.

The new sentences “One should be noted that, in theory, HIF is widely accepted to normally occur in the $ABO_3/A'BO_3$ perovskite superlattices rather than $ABO_3/AB'O_3$ ”

systems^{9, 11, 53}. Since A-site and B-site positions have different inversion symmetry, the octahedral rotations combined with A/A' layered cation ordering can induce effectively polar trilinear term, i.e., the chemical criterion; the rotation pattern of $a^-a^+c^+$ is necessary to dominate the energy landscape over other competing instabilities and drive the transition to the polar structure, i.e., energy criterion⁵³. For LNMO/LCMO superlattices with B-site cation orderings, it seems to be contrary to the chemical criterion of HIF. However, this double-perovskite superlattice with $a^-a^+c^+$ rotations is much more different from the general systems of $ABO_3/AB'O_3$. Under epitaxial strain, various oxygen octahedrons (MnO_6 , CoO_6 and NiO_6) will develop different degrees of rotation/tilting and JT distortion due to the different chemical properties, resulting in the different inversion symmetry of B-site positions. Furthermore, the multimodal epitaxial growth provides a relatively complex lattice environment, driving a polar trilinear term similar to that in the A/A' ordered system. Therefore, in fact, many experimental factors can cause the breaking of inversion symmetry to some extent or locally in perovskite superlattices system, such as lattice distortions, oxygen vacancies, dislocations, strain and cation orderings.

Moreover, the B-site ordered perovskite superlattice systems have been increasingly focused on the theoretical study of hybrid ferroelectricity and multiferroics. The coexistence of lattice distortions and charge transfer can induce the hybrid ferroelectricity or polar structures in the perovskite superlattices $(ABO_3)_n/(AB'O_3)_n$ ($n=2$)^{54, 55}. The combination of the ferroelectricity induced by octahedral rotation and the ferromagnetism/ferrimagnetism caused by the ordered arrangement of different magnetic ions is also calculated in the B-site ordered double-perovskite bilayer¹⁰.” were added in Discussion (page 19/line 15 to page 20/line 10).

The new sentences “It seems that the double-perovskite superlattice system of $(A_2BB'O_6)_n/(A_2BB''O_6)_n$ ($n=1$) could be a novel platform for achieving HIF through the unique structural design.” were added in page 20/lines 21-23.

Added References:

53. J. M. Rondinelli & C. J. Fennie. Octahedral rotation-induced ferroelectricity in cation ordered perovskites. *Adv. Mater.* **24**, 1961-1968 (2012).

54. H. Zhang, Y. Weng, X. Yao & S. Dong. Charge transfer and hybrid ferroelectricity in \$(\text{YFeO}_3)_n/(\text{YTiO}_3)_n\$ magnetic superlattices. *Phys. Rev. B* **91**, 195148 (2015).
55. Y. Weng, J. Zhang, B. Gao & S. Dong. \$(\text{LaTiO}_3)_n/(\text{LaVO}_3)_n\$ as a model system for unconventional charge transfer and polar metallicity. *Phys. Rev. B* **95**, 155117 (2017).

Comment 3: *The tool/method of simulating the oxygen positions should be added in the part of methods in the article.*

Response: Thanks for the reviewer's valuable comments. According to this comment, we have added the related details of simulating the oxygen positions for the determinations of BO_6 octahedral deformation and OOR/OOT to Methods in the manuscript.

We build a crystal cell of LNMO/LCMO double-perovskite superlattices with boundary symmetry conditions for DFT and simulate the structural changes by applying equivalent strain (Fig. 4c and Supplementary Fig. S31). The applied in-plane and out-of-plane strains were equivalently replaced by the changes of lattice parameters in the experiment (Fig. 1e). Based on the optimization results of the strained crystal structures, the oxygen positions can be determined, thus the BO_6 octahedral deformation and OOR/OOT are further determined by measuring the $B\text{-O}$ bond lengths and $B\text{-O-B}$ bond angles, respectively (Supplementary Table S2 and S3).

Thanks again for the reviewer's useful suggestion. The corresponding revisions clearly illustrate the simulation method for octahedral distortions and improve the readability of the manuscript.

Revision:

The new sentences "A crystal cell of LNMO/LCMO double-perovskite superlattices with boundary symmetry conditions was built for DFT calculation. The applied IP and OP strains were equivalently replaced by the changed lattice parameters in the

experiment. Based on the optimization results of the strained crystal structures, the oxygen positions can be determined, and the BO_6 octahedral deformation and OOR/OOT are further determined by measuring the $B-O$ bond lengths and $B-O-B$ bond angles, respectively.” were added in Methods (page 22/line 29 to page 23/line 5).

Finally, thanks to the editor and reviewers again for all the excellent and professional comments, which give us a great chance to improve our work better.

REVIEWERS' COMMENTS

Reviewer #2 (Remarks to the Author):

The authors have responded to all the concerns well and revised carefully. It can be accepted as it is.

Response to Reviewers' Comments

Reviewer #2 (Remarks to the Author):

Comments: *The authors have responded to all the concerns well and revised carefully. It can be accepted as it is..*

Response: Thanks for the reviewer's positive comments. We appreciate the reviewers' time and effort again for improving our manuscript.